# Path-seq identifies an essential mycolate remodeling program for mycobacterial host adaptation

Eliza JR Peterson[1] ID, Rebeca Bailo[2] ID, Alissa C Rothchild[3] ID, Mario L Arrieta-Ortiz[1], Amardeep Kaur[1], Min Pan[1], Dat Mai[3], Abrar A Abidi[1], Charlotte Cooper[2], Alan Aderem[3], Apoorva Bhatt[2,*] ID & Nitin S Baliga[1,4,5,**] ID

## Abstract

The success of *Mycobacterium tuberculosis* (MTB) stems from its ability to remain hidden from the immune system within macrophages. Here, we report a new technology (Path-seq) to sequence miniscule amounts of MTB transcripts within up to million-fold excess host RNA. Using Path-seq and regulatory network analyses, we have discovered a novel transcriptional program for *in vivo* mycobacterial cell wall remodeling when the pathogen infects alveolar macrophages in mice. We have discovered that MadR transcriptionally modulates two mycolic acid desaturases *desA1/desA2* to initially promote cell wall remodeling upon *in vitro* macrophage infection and, subsequently, reduces mycolate biosynthesis upon entering dormancy. We demonstrate that disrupting MadR program is lethal to diverse mycobacteria making this evolutionarily conserved regulator a prime antitubercular target for both early and late stages of infection.

**Keywords** gene regulatory networks; host–pathogen interactions; *Mycobacterium tuberculosis*; Path-seq; systems biology
**Subject Categories** Genome-Scale & Integrative Biology; Microbiology, Virology & Host Pathogen Interaction
**Mol Syst Biol. (2019) 15: e8584**

## Introduction

*Mycobacterium tuberculosis* (MTB) infection occurs by inhalation of bacilli-containing aerosols. Alveolar macrophages, which line the airway, are the first host cells to phagocytize the bacteria. This initial contact of MTB with alveolar macrophages begins a complex battle between bacterial virulence and host immunity, orchestrated in large part by intricate gene regulatory pathways (Galan & Wolf-Watz, 2006; Medzhitov, 2007). As such, measuring gene expression *in vivo* is central to our understanding of TB disease control and progression (Flynn *et al*, 2011).

RNA-seq provides a sensitive method for global gene expression analysis. Specific for infection biology, dual RNA-seq methods have allowed simultaneous profiling of host and pathogen RNA. However, the striking excess of eukaryotic over bacterial RNA limits the coverage of pathogen transcripts in dual RNA-seq studies (Avraham *et al*, 2015, 2016, Rienksma *et al*, 2015; Westermann *et al*, 2012; Westermann *et al*, 2016), and methods to partially enrich for bacterial transcripts have had limited success (Humphrys *et al*, 2013; Mavromatis *et al*, 2015). It is clear that more sensitive approaches are needed to profile the transcriptional state of the pathogen during infection, especially *in vivo*.

To improve the coverage of pathogen transcripts, we made use of biotinylated oligonucleotide baits that are complementary to the pathogen transcriptome. The baits are hybridized to mixed host–pathogen RNA and used to enrich pathogen transcripts for sequencing. Approaches using biotinylated genome fragments have previously been used to enrich specific transcripts of intracellular pathogens (Graham & Clark-Curtiss, 1999; Morrow *et al*, 1999) or perform genome-wide transcriptome profiling of fungal RNA from infected host cells (Amorim-Vaz *et al*, 2015). Here, we applied our pathogen-sequencing (Path-seq) method to explore transcriptional changes in MTB (one-fourth the size of fungus, *Candida albicans*) following infection in mice. Path-seq data along with network modeling have led to discovery that MTB transcriptionally regulates mycolic acids during infection of host cells, influencing virulence and persistence of the pathogen.

1 Institute for Systems Biology, Seattle, WA, USA
2 School of Biosciences and Institute of Microbiology and Infection, University of Birmingham, Birmingham, UK
3 Center for Global Infectious Disease Research, Seattle Children's Research Institute, Seattle, WA, USA
4 Molecular and Cellular Biology Program, Departments of Microbiology and Biology, University of Washington, Seattle, WA, USA
5 Lawrence Berkeley National Laboratories, Berkeley, CA, USA
*Corresponding author. Tel: +44 121 41 45893; E-mail: a.bhatt@bham.ac.uk
**Corresponding author. Tel: +1 2067321266; E-mail: nitin.baliga@systemsbiology.org

# Results

### Development of Path-seq

To enrich the bacterial pathogen transcripts, we used Agilent eArray (Ong *et al*, 2011) to create a custom bait library that covers all MTB transcripts at even intervals. Our MTB library contains 35,624 probes, each with biotinylated oligonucleotides of 120 base lengths. The bait library composition is modular and can be designed to cover specific transcripts of interest. Similarly, transcripts such as rRNA can be excluded or gene sequences altered for polymorphisms found in clinical strains (Fleischmann *et al*, 2002). For this study, we chose all transcripts of MTB H37Rv for complete coverage and comparison with standard RNA-seq results.

To assess enrichment of pathogen transcripts, we first used RNA isolated from murine bone marrow-derived macrophages (BMDMs) spiked with 0.1% MTB RNA. A typical mammalian cell contains on the order of 20 picograms of RNA, which is roughly two orders of magnitude more than a single bacterial cell (Alberts *et al*, 1994). Accounting for BMDMs that might not be infected and based on intracellular sequencing studies from literature (Avraham *et al*, 2016; Westermann et al, 2016), we estimated 0.1% pathogen RNA would be representative of a typical *in vitro* infection. We performed double rRNA depletion using Illumina Ribo-Zero Gold Epidemiology Kit and used the SureSelect protocol to generate strand-specific libraries for sequencing. Half of the library was then indexed for sequencing as the "RNA-seq" sample, and the other was hybridized to the probes, amplified, and indexed as the "Path-seq" sample (Fig 1A). We performed three replicate experiments of the mock infection using the same MTB RNA. With the probe hybridization, the percentages of reads aligned to MTB were increased up to

840-fold. Both the normalized read counts (Fig 1B) and enrichment efficiency (inset Fig 1B) were highly reproducible across three replicate samples. Repeating the Path-seq method with spiked RNA samples, we increased the proportion of macrophage RNA and were able to quantify MTB transcripts from one millionth of the host RNA (1.75% of all reads aligned to MTB genomes).

To validate the enrichment protocol yielded quantitatively reliable read counts, we investigated the correlation between the RPKMs obtained from the sequencing of RNA from *in vitro* grown MTB, without enrichment (RNA-seq), and the RPKM values obtained with the enrichment protocol (Path-seq) using the same 0.1% MTB RNA with host RNA (BMDMs). Even using different library preparation kits (Illumina for RNA-seq and Agilent for Path-seq), the correlation of RPKMs was 0.92–0.93 (Fig 1C), demonstrating the enrichment process was efficient and accurate for gene expression analysis.

### Analysis of MTB transcriptome during *in vivo* infection using Path-seq method

Little is known about the transcriptional state of the pathogen during infection of animal models (Talaat *et al*, 2004); technical challenges have limited these studies. Given the enrichment capabilities of the Path-seq method, we evaluated the use of the approach to study the transcriptome of MTB isolated from alveolar macrophages (AMs) of infected mice. We used fluorescence-activated cell sorting (FACS) and gating strategies to isolate AMs from bronchoalveolar lavage (BAL) of mice at 24 h post-aerosol infection with MTB. We isolated AMs from BAL, instead of whole lung tissue, to avoid the harsh digestion step at 37°C that can alter the transcriptional state of the cells. RNA extracted from AMs in BAL of 10 mice yielded

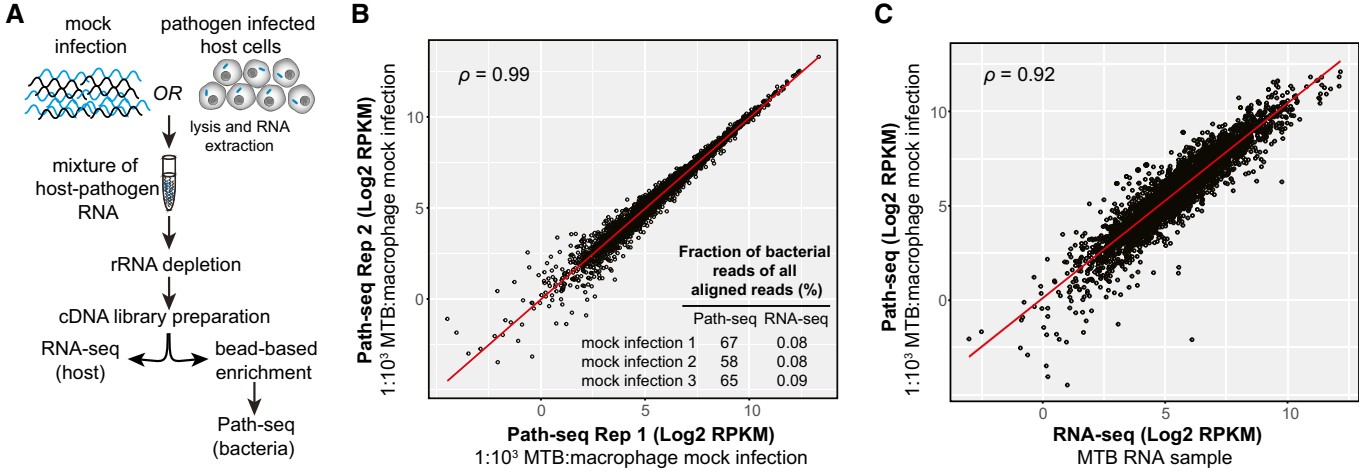

**Figure 1. Path-seq workflow and validation.**

A  Total RNA from mock infection or infected cells was depleted of rRNA, and cDNA libraries were prepared. Libraries were then either indexed and sequenced directly for host transcripts or enriched using pathogen-specific oligonucleotides bound to beads. After hybridization, enriched libraries were indexed, sequenced, and reads assigned to host or pathogen genomes *in silico*.

B  Correlation between replicate mock infections. Path-seq reads were recovered from samples of macrophage RNA spiked with 0.1% MTB RNA. Scatter plot of log2 RPKM values is shown with Pearson correlation, *P*-value < 0.0001. Inset summarizes the mock infection replicates and their fraction of MTB reads (of all aligned reads) from Path-seq and standard RNA-seq methods.

C  Correlation between MTB RNA sequenced by RNA-seq (Illumina TruSeq Library Prep) and the same MTB RNA sample combined with macrophage RNA at 1:1,000 ratio and processed using Path-seq method. Scatter plot of log2 RPKM values is shown with Pearson correlation, *P*-value < 0.0001.

~100 μg of total RNA. Therefore, we first evaluated the Path-seq method using 0.3 μg of BMDM RNA spiked with 0.005% MTB RNA, to simulate mixed host and pathogen RNA composition of a sample from an *in vivo* infection. We performed Path-seq with two replicates, and alignment analysis revealed the percentages of reads that aligned to MTB were 38 and 27%, an approximate 10,000-fold enrichment.

After evaluating the Path-seq methods' feasibility for *in vivo* MTB transcriptome analysis, we used flow cytometry to isolate AMs (average of 4.3% of all cells and 83.1% of live, CD45+ cells) in BAL of 30 mice 24 h after infection with wild-type MTB (Appendix Fig S1A). Infection, FACS sorting (Appendix Fig S1B), and RNA extraction were repeated with three independent mouse infections (three biological replicates), yielding an average of ~300 μg total RNA per replicate. The Path-seq enrichment was performed and resulted in 17, 8, and 5% of the entire reads aligning to MTB from each of the replicates. We compared the MTB read counts between the *in vivo* samples with extracellular samples, biological replicates of RNA extracted from MTB grown in 7H9 media for 24 h (starting $OD_{600} = 0.1$). Both the *in vivo* and extracellular samples were processed by Path-seq. While the percentage of non-zero reads and total read counts are lower in the *in vivo* samples, the mean count per gene and coefficient of variation are the same between the two conditions (Appendix Table S1). This gives us confidence that for genes with detectable reads, we are measuring real expression levels. We suspect genes with non-detectable reads are a result of the miniscule amount of MTB RNA compared to host RNA in the *in vivo* samples, and not a reflection of real gene expression changes. Therefore, excluding genes with zero counts in all *in vivo* replicates resulted in 3,505 MTB genes (62% of genome) with sequenced expression measurements from *in vivo* infection using Path-seq. These results (raw data and normalized read counts for ALL genes are available in GEO: GSE116394) present the most comprehensive transcriptome profiling of MTB from *in vivo* infection and a major technical advancement for researchers studying host–pathogen interactions.

To calculate differentially expressed genes between the *in vivo* and extracellular samples, we excluded *in vivo* biological replicate 3, which had significantly lower total read counts compared to all other samples (summarized in Appendix Table S1). The lower deposition of replicate 3 suggests a lower amount of starting MTB could be the reason for the low read count. Differential expression analysis between *in vivo* intracellular MTB and extracellular MTB identified 431 significantly differentially expressed transcripts (log2 fold change $< -1.0$ or $> 1.0$ and multiple hypothesis-corrected $P$-value $< 0.05$, Dataset EV1).

Among the differentially expressed transcripts that code for annotated proteins (376 genes), 121 were down-regulated and significantly enriched (multiple hypothesis-corrected $P$-value = 0.005), in the Mycobrowser category (Kapopoulou *et al*, 2011) of proline–glutamic acid (PE)/proline–proline–glutamic acid (PPE) family of proteins. The exact physiological role of the PE and PPE proteins in MTB is yet to be fully understood, but they are thought to play important roles in immune evasion (Tiwari *et al*, 2012). It is interesting that PE and PPE genes were down-regulated in AMs and might indicate that they are unnecessary within these host cells. In addition, 255 genes were up-regulated in AMs at 24 h post-infection

and were significantly enriched ($P < 0.05$) in the Mycobrowser functional categories: "insertion sequences and phages", "information pathways", and "lipid metabolism". Most interesting, many of the genes whose protein products are associated with "lipid metabolism" are involved in the biosynthesis of mycolic acid. These up-regulated mycolic acid biosynthesis genes included *umaA, pcaA, desA1, desA2, fadD32,* and *fabD.* In addition, genes of the operon involved in phthiocerol dimycocerosate (PDIM) biosynthesis were also up-regulated. The biosynthesis of new cell wall material is an energetically expensive process and found to be repressed in MTB upon entry into dormancy (Galagan *et al*, 2013; Jamet *et al*, 2015). This suggests that MTB in AMs 24 h post-infection are not in a dormant state. Instead, the transcriptional response indicates MTB is actively remodeling the cell wall, perhaps with modifications that specifically contribute to survival within AMs. Interestingly, the up-regulated genes, *umaA* and *pcaA*, are required for cyclopropane ring formation in mycolic acids of MTB. Furthermore, *desA1* and *desA2* (with log2 fold change of 4.0 and 4.7, respectively, within AMs) encode fatty acid desaturases that introduce double bonds into fatty acids (Singh *et al*, 2016). Desaturation is a necessary step prior to cyclopropanation and other mycolic acid modifications. It is interesting to speculate that conditions within AMs induce desaturation events, enabling MTB to fine-tune subsequent cyclopropanation and other modifications of mycolic acids that contribute to cell wall permeability and adaptation within these host cells. The significant up-regulation of mycolic acid remodeling genes following *in vivo* infection was interesting and deserved further investigation of their transcriptional control.

## Genome-wide expression analysis during *in vitro* macrophage infection using Path-seq

Several genome-wide expression studies of MTB challenged with dormancy-inducing stresses, such as nutrient starvation (Jamet *et al*, 2015) and hypoxia (Galagan *et al*, 2013; McGillivray *et al*, 2015), have shown that genes involved in mycolic acid biosynthesis are generally down-regulated. In contrast, we observed up-regulation of mycolate biosynthesis genes in MTB from AMs of infected mice at 24 h. Therefore, we sought to study the expression of these mycolic acid modification genes at multiple time points during infection using the Path-seq method and MTB-infected bone marrow-derived macrophages (BMDMs). An *in vitro* infection system was used due to the large number of mice required for additional time points during *in vivo* infection. We isolated murine BMDMs and infected them with MTB at a MOI of 10. Infected cells were collected at 2, 8, and 24 h after infection along with extracellular MTB grown in 7H9 media as control. Total RNA was extracted, depleted of rRNA, and handled as described above (Fig 1A). All extracellular MTB samples were processed by Path-seq as well. For the *in vitro* infection samples (Appendix Fig S2), we split each sample into RNA-seq and Path-seq fractions to evaluate the enrichment efficiency and to simultaneously obtain both host and pathogen transcriptomes from the same infection sample. While we did not perform transcriptome analysis of the host cells in this study, the raw data are available (GSE116357) along with uninfected BMDM controls and represents the first duel monitoring of both MTB and host transcriptomes from the same infection samples. The percentage of reads that aligned to MTB was consistent at a 100-fold

increase in the enriched vs non-enriched samples across replicates and time points (Table 1). With an average of 11 million (M) mapped reads for both intracellular (average 13.4 M) and extracellular (average 8.8 M) MTB, we obtained >100× coverage and 5,622 unique features (including ncRNA and UTRs). This is further validation of the Path-seq method to comprehensively study the authentic intracellular state of a pathogen.

Using the normalized read counts from the intracellular and extracellular MTB data, we identified two clusters by implementing the R NbClust function (Charrad *et al*, 2014) on principal component analysis output, a dimensionality reduction method. The two identified clusters are shown in a two-dimensional *t*-distributed stochastic neighbor embedding plot (*t*-SNE, van der Maaten & Hinton, 2008) plot. Extracellular samples clustered closely together, distinct from the intracellular samples and according to their time post-infection (Appendix Fig S3A). Biological replicates fell into related groups and demonstrated strong correlation in pairwise comparison of RPKMs (Appendix Fig S3B). Differential expression of intracellular MTB was calculated relative to extracellular, at each time point using DESeq2. Overall, there were 746, 945, and 412 significant differentially expressed (log2 fold change < −1.0 or > 1.0 and multiple hypothesis-adjusted *P*-value < 0.01) transcripts at the 2, 8, and 24 h post-infection time points, respectively (Dataset EV2). The most up-regulated genes at all time points included genes such as *icl1*, *Rv1129c*, *prpD*, *prpC*, and *fadD19*. The induced expression of these genes is consistent with known alterations in lipid degradation during infection, enhanced activity of the methylcitrate cycle (Munoz-Elias *et al*, 2006), and genetic evidence that MTB utilizes cholesterol from the host during infection (Pandey & Sassetti, 2008). These carbon-metabolizing genes were also found to be up-regulated in microarray analysis of *in vitro* MTB-infected host cells (Schnappinger *et al*, 2003; Data ref: Schnappinger *et al*, 2003; Rohde *et al*, 2007; Data ref: Rohde *et al*, 2007), along with a significant overlap of other differentially expressed genes between the datasets (Appendix Table S2). These data demonstrate that the Path-seq method yielded data consistent with published transcriptional studies of *in vitro* infected host cells. Importantly, the Path-seq method allows for simultaneous expression profiling of host and pathogen transcripts and additional transcript features that are not possible in microarray studies.

### desA1 and desA2 are induced early during *in vitro* macrophage infection and hypoxia time course

Among the mycolic acid biosynthesis genes, only *umaA* was up-regulated at all time points and *desA1*/*desA*2 were transiently up-regulated at 2 h following MTB infection of BMDMs. Similarly, *umaA*, *desA1*, and *desA2* were also found to be up-regulated in the *in vitro* infection microarray analyses (Schnappinger *et al*, 2003; Data ref: Schnappinger *et al*, 2003; Rohde *et al*, 2007; Data ref: Rohde *et al*, 2007), but none of the other mycolic acid biosynthesis genes that were up-regulated *in vivo*. In the *in vitro* infection using Path-seq, after 2 h, *desA1*/*desA2* returned to levels similar to extracellular MTB at 8 h and 24 h (Fig 2A). This expression dynamics of *desA1*/*desA2* during MTB infection of BMDMs was also mirrored in RNA-seq data of MTB entering and exiting hypoxia over a 5-day time course (Fig 2B). In this experiment, we used mass flow controllers to regulate the amount of air and nitrogen ($N_2$) gas streaming into cultures of MTB and achieve a gradual depletion of oxygen over 2 days. The cultures were maintained in hypoxia for 2 days by streaming only $N_2$ and then reaerated over 1 day by a controlled increase in air flow. During the 2-day oxygen depletion, the expression levels of *desA1* and *desA2* did not change significantly. However, as soon as the cultures reached complete hypoxia (0% dissolved oxygen), the expression of the desaturases increased for ~5 h, followed by a dramatic repression after ~30 h of being in hypoxia. Subsequently, reaeration of the culture returned *desA1* and *desA2* to basal expression levels (Fig 2B). Interestingly, *umaA* was not significantly differentially expressed across the hypoxia time course.

### Gene expression comparison between *in vivo* and *in vitro* infections

In addition to the mycolic acid biosynthesis gene, we further compared all significantly differentially expressed genes between the two infection models and found only a small but significant (*P*-value < 0.01) subset of common genes (59 genes at 24 h and 137 genes from any *in vitro* time point, Dataset EV3 and Appendix Fig S4). Most of the common genes were up-regulated in both models and included genes significantly enriched in categories related to the ribosome and response to hypoxia according to MTB annotation in DAVID (Huang da *et al*, 2009a,b). Interestingly, both AAA$^+$ ATPases, *clpX* (Rv2457c), and *clpC1* (Rv3596c) that interact with the ClpP proteolytic core (Neuwald *et al*, 1999; Raju *et al*, 2014) were also significantly up-regulated in both models. Despite these few similarly expressed genes, the strikingly different gene expression profiles between the experimental infection models could reflect heterogeneity in host cells. Huang *et al* (2018) recently demonstrated in mice that MTB has lower bacterial stress in AMs compared to interstitial macrophages (IMs) at 2 weeks post-infection. The authors theorize that different host macrophage lineages represent different intracellular environments that are permissive (AMs) or restrictive (IMs) for MTB growth (Huang *et al*, 2018). Our data also point toward differences in the MTB transcriptional state from macrophages of different lineages.

**Table 1. Genome mapping statistics from Path-seq and RNA-seq from *in vitro* infection of MTB-infected BMDMs.**

| | Fraction of MTB reads of all aligned reads (%) | |
| --- | --- | --- |
| | **Path-seq** | **RNA-seq** |
| Intracellular 2 h: a | 77.0 | 0.73 |
| Intracellular 2 h: b | 71.1 | 0.53 |
| Intracellular 2 h: c | 88.5 | 0.65 |
| Intracellular 8 h: a | 47.7 | 0.54 |
| Intracellular 8 h: b | 87.5 | 0.66 |
| Intracellular 8 h: c | 82.8 | 0.38 |
| Intracellular 24 h: a | 97.4 | 3.97 |
| Intracellular 24 h: b | 75.5 | 1.88 |
| Intracellular 24 h: c | 97.5 | 3.10 |

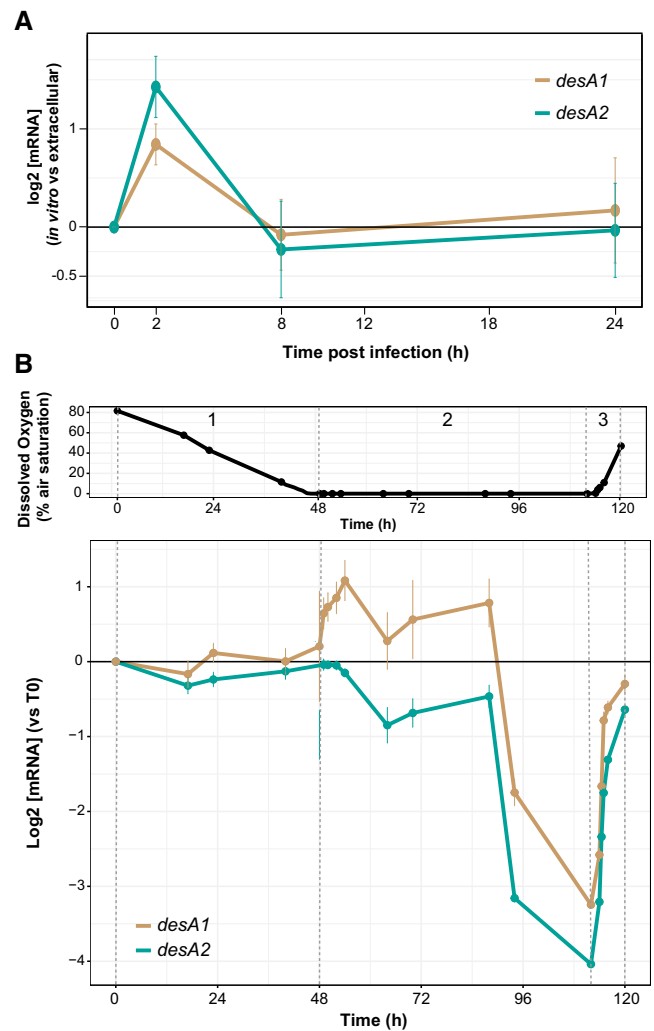

**Figure 2.    The expression levels of *desA1* and *desA2* from Path-seq of MTB-infected BMDMs and over a hypoxia time course.**

A    Path-seq profiles of *desA1* and *desA2* in MTB-infected BMDMs. Error bars show the standard deviation from three biological samples. Representative results from two experiments are presented.

B    Upper panel: dissolved oxygen curve over 120-h time course showing controlled depletion (1), sustained hypoxia (2), and controlled reaeration (3). Points represent the average of three biological replicates and were measured via fiber optic technology that non-invasively probes oxygen levels in the culture (PreSens Precision Sensing GmbH). Lower panel: expression profiles (RNA-seq) of *desA1* and *desA2* over the time course and oxygen levels. Error bars show the standard deviation from three biological samples.

## Systems-level comparison of active MTB regulatory networks illustrates differences between infection models

Given the transcriptome differences between the infection models, we employed a computational framework to characterize, at systems scale, the transcriptional differences between the extracellular and intracellular states for each model. There are numerous methods for identifying the key transcriptional networks from different environments (Balazsi *et al*, 2008; Brynildsen & Liao, 2009; Cahan *et al*, 2014). Based on one such approach, NetSurgeon (Michael

*et al*, 2016), we evaluated the role of each TF in the observed gene expression changes given a signed transcriptional network. We constructed a transcriptional network based on ChIP-seq data from overexpression of 178 of 214 TFs in MTB (Minch *et al*, 2014; Data ref: Minch *et al*, 2015). Activating and repressing influences of TFs were inferred from consequence of TF overexpression on downstream genes (Rustad *et al*, 2014; Data ref: Rustad *et al*, 2014). Using a data-driven transcriptional network of 4,635 interactions, each TF–target gene interaction was weighted according to the multiple hypothesis-adjusted *P*-value from differential expression analysis between intracellular and extracellular conditions. We calculated a relative score for each TF in conditions simulating deletion or overexpression of the TF. These simulations prioritized TF activities (decreased or increased) yielding a transcriptome most similar to the infected state, compared to the control (see Materials and Methods and summary schematic in Fig 3A). We performed this analysis for each time point and infection model to identify highly ranked TFs (Fig 3B).

From the *in vitro* macrophage infection, many of the TFs had distinct temporal activity, while others were highly ranked across all time points (Fig 3B). These sustained regulons include DosR, which is known to contain a set of ~50 genes that are induced in response to multiple signals including hypoxia, nitrosative stress, and carbon monoxide (Park *et al*, 2003; Kendall *et al*, 2004; Roberts *et al*, 2004; Kumar *et al*, 2008). While DosR regulon induction is typically associated with hypoxic conditions and reactive nitrogen intermediates (RNIs), we observed activation as early as 2 h post-infection. Encouragingly, this 2 h induction was also found in the microarray study of MTB-infected macrophages, where high DosR regulon expression was sustained until a striking down-regulation at Day 8 (Rohde *et al*, 2012). This indicates that the known cues of this regulatory network are present almost immediately during *in vitro* infection. In addition to DosR, two other TFs had high activity across all time points, Rv0681 (Fig 3C) and Rv0691c. Interestingly, both are TetR family transcriptional regulators and conserved across all mycobacterial genomes (Balhana *et al*, 2015), including the drastically reduced *Mycobacterium leprae*. The function of these transcriptional networks is unknown, but suggests their activity is important for survival in both environmental and intracellular niches.

Among the TFs with decreased activity, KstR and KstR2 were found across all time points of the *in vitro* infection and are known to repress genes required for cholesterol utilization (Kendall *et al*, 2010). Our analysis indicates that reduction of their repressive activity, and increased expression of their target genes, is important for driving the *in vitro* intracellular transcriptional state. This is consistent with the highly expressed cholesterol utilization and methyl citrate cycle genes that we and others have observed (Schnappinger *et al*, 2003; Rohde *et al*, 2007; Homolka *et al*, 2010). Moreover, this emphasizes the importance of altered carbon metabolism and utilization of host-derived nutrients as key to MTB *in vitro* intracellular adaptation. Another repressor, Zur (previously FurB), had decreased activity across all time points (Fig 3D). Zur down-regulates genes involved in zinc transport (Maciag *et al*, 2007). During MTB infection, macrophages overload the phagosome with copper and zinc as a strategy to poison the pathogen (Neyrolles *et al*, 2015). However, through multi-faceted resistance mechanisms we do not fully appreciate, MTB is able to protect itself against metal toxicity. Our analysis proposes that reduced Zur activity results in increased

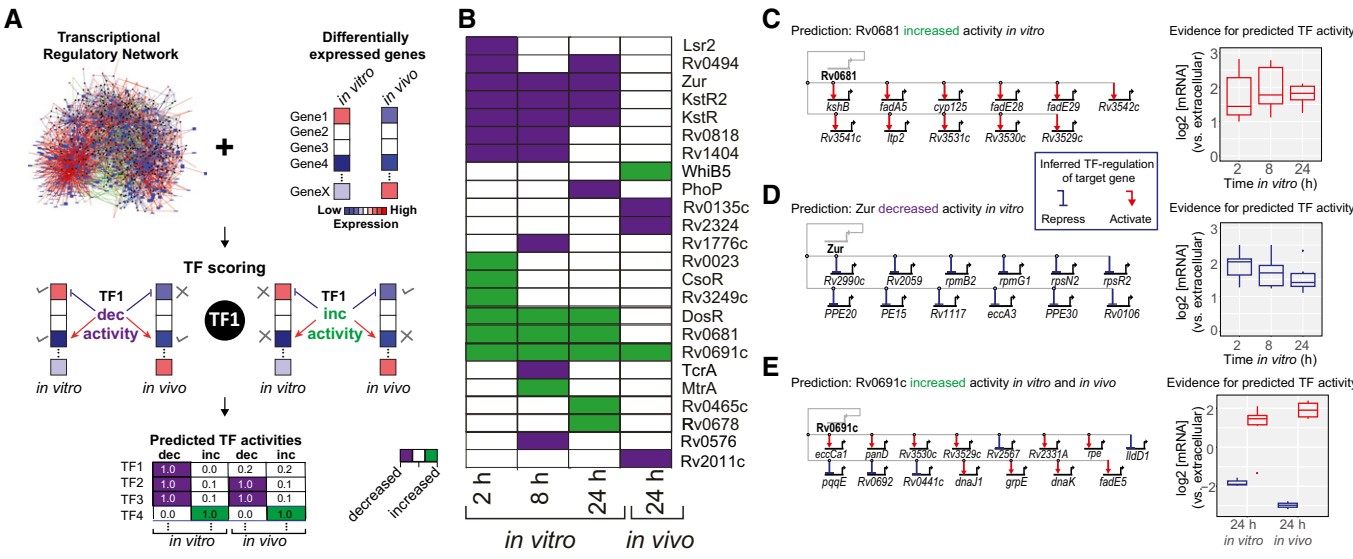

**Figure 3. Systems approach to identify active intracellular regulatory networks.**

A  Schematic of network analysis to identify TFs with activity (increased or decreased) in controlling the transcriptional state of MTB during infection of host cells. Abbreviations: dec; decreased, inc; increased.

B  Heatmap of TFs with decreased (purple) or increased (green) activity at specific time points during *in vitro* or *in vivo* infection.

C  Rv0681 regulon genes differentially expressed *in vitro* and the evidence for predicted high Rv0681 activity (induced expression of target genes).

D  Zur regulon genes differentially expressed *in vitro* and the evidence for decreased Zur activity (derepresses target genes).

E  Rv0691c regulon genes differentially expressed *in vitro* and *in vivo*; evidence for increased Rv0691c activity (increased up- and down-regulation of target genes) at 24 h from both *in vitro* and *in vivo* infection.

Data information: (C–E) Boxplots represent log2 fold change (FC) values of TF target genes. The boxes show the median and first/third quartiles of the log2 FC values; the whiskers extend to the smallest/largest values that are no further than 1.5 times the inter-quartile range.

---

expression of zinc transport genes which could help with regulating zinc levels in MTB during *in vitro* macrophage infection. Interestingly, other regulators of metal content (TFs, uptake and export) were recently found to be required for *in vitro* intracellular growth by high-content imaging of an MTB transposon mutant library (Barczak *et al*, 2017). Leveraging our Path-seq data, we developed a systems-level approach that recapitulates known *in vitro* intracellular regulatory networks and prioritizes others for further experimental testing.

We also applied the same analysis to the *in vivo* expression data (using differentially expressed genes in Dataset EV1) to identify transcriptional networks involved in MTBs response within AMs. Interestingly, we observed very few networks that were active in both infection models. Only Rv0691c was highly ranked at 24 h from both AMs and BMDMs (Fig 3E). In our regulatory network, Rv0691c has ~50 target genes, a subset of which are up- and down-regulated during *in vitro* and *in vivo* infection. The genes in the regulon do not categorize into a certain pathway, but our unbiased analysis suggests the Rv0691c regulon deserves further study for its role in establishing MTB infection both *in vitro* and *in vivo*. Overall, there were far fewer active networks identified *in vivo*, compared to the *in vitro* infection. While the type of differentially expressed genes (i.e., genes not belonging to regulons) could contribute to such differences, we do not see that being the case. Therefore, it is appealing to speculate that the more permissive environment within AMs or a greater heterogeneity of infection from the *in vivo* model could contribute to the differences in the number of active regulatory networks identified by NetSurgeon.

### Identification of EGRIN module relevant to infection and *desA1/desA2* transcriptional regulator, Rv0472c

Our systems analysis revealed novel and infection-specific regulatory networks. However, none of the identified regulons included the mycolic acid biosynthesis genes up-regulated *in vivo*, and particularly *umaA* and *desA1/desA2* that were up-regulated in both infection models. One plausible explanation could be the regulon size threshold that was implemented to reduce false positives (TFs with at least five targets were considered in this analysis). Therefore, we used an orthogonal network approach to discover regulatory mechanisms controlling the expression of these genes that could be important for mycolic acid remodeling during infection. Specifically, we used the environment and gene regulatory influence network (EGRIN) model of MTB that was previously published and demonstrated to accurately predict regulatory interactions through validation with the DNA binding sites and transcriptional targets from overexpressing > 150 MTB transcription factors (TFs; Peterson *et al*, 2015; Turkarslan *et al*, 2015). The full description of the algorithms used to construct the EGRIN model is beyond the scope of this work; readers are encouraged to refer to the original paper for more detail (Reiss *et al*, 2006). Briefly, the EGRIN model was constructed through semi-supervised biclustering of a compendium of 2,325 transcriptomes assayed during MTB response to diverse environmental challenges, guided by biologically informative priors and *de novo* cis-regulatory GRE detection for module (also referred to as bicluster) assignment. Overall, the EGRIN model is sufficiently predictive to formulate hypotheses of MTB regulatory interactions that

respond to various environmental conditions, including new conditions not represented in the gene expression compendium such as *in vivo* infection. Using the EGRIN model, we identified significant enrichment (multiple hypothesis-corrected *P*-value = $4.9 \times 10^{-9}$) of the *in vivo* up-regulated mycolic acid biosynthesis genes in bicluster 276 and with predicted regulation by the TF, Rv0472c (Fig 4A). Bicluster 276 contains the desaturases, *desA1/desA2*, and PDIM synthases, *ppsD/E*, all of which were up-regulated *in vivo*. Furthermore, bicluster 276 contains the toxin–antitoxin system, *vapB/C47*, which was also significantly up-regulated *in vivo* (with log2 fold change of 3.5 and 3.0, respectively). The PDIM biosynthesis and transport genes, *fad26*, *drrABC,* and *papA5*, are additional gene members of module 276. Interestingly, all of the PDIM-related genes of module 276 genes (*ppsD/E*, *drrABC*, *papA5*) were identified in a high-content imaging analysis of bacterial mutants during macrophage infection (Barczak *et al*, 2017). More specifically, they found mutants of these genes impaired intracellular survival and reduced type I interferon (IFN) response in host cells (Barczak *et al*, 2017). Although the detrimental versus beneficial relevance of type I IFN in MTB infection remains a matter of active debate, growing evidence suggests type I IFN promotes bacterial expansion and pathogenesis within host cells (Moreira-Teixeira *et al*, 2018). As such, we presume module 276 genes are up-regulated upon *in vivo* infection to collectively alter cell wall composition and modulate the immune system, thereby promoting MTB survival and proliferation within AMs.

The EGRIN model predicted regulation of module 276 by a TetR-type TF, Rv0472c, with homologs across all mycobacteria, including *M. leprae* (Balhana *et al*, 2015). When overexpressed in MTB, Rv0472c led to significant repression of 15 genes, but only *desA1* and *desA2* had significant binding of Rv0472c in their promoter region from ChIP-seq analysis (Minch *et al*, 2014; Data ref: Minch *et al*, 2015). Given the conservation of Rv0472c across mycobacteria, we hypothesized that overexpression of the MSM homolog should also repress the desaturases in MSM (Fig 4A). We cloned *MSMEG_0916* into an anhydrotetracycline (ATc)-inducible Gateway shuttle vector as previously described for MTB (Galagan *et al*, 2013; Data ref: Minch *et al*, 2015) and transformed into MSM. We induced expression of *MSMEG_0916* for 4 h and harvested chromatin samples for ChIP-seq as well as RNA for transcriptional profiling by RNA-seq. Overexpression of *MSMEG_0916* resulted in nine significant ChIP peaks (*P*-value < 0.01) with a peak score higher than 0.7, as analyzed by DuffyNGS ChIP peak calling method (see Methods). Among these were peaks located in the promoter of the MSM *desA1* and *desA2* (Fig 4B). Additionally, *MSMEG_0916* overexpression resulted in significant repression of *desA1* and *desA2*, with a log2 fold change of −1.32 and −1.72, respectively, compared to uninduced (Fig 4C). The DNA consensus motifs, generated using MEME and DNA binding data from ChIP-seq, also had significant alignment between Rv0472c and *MSMEG_0916* (Appendix Fig S5).

This analysis demonstrates the utility of EGRIN to identify the regulator of genes relevant to infection. Yet, the *Rv0472c* and *MSMEG_0916* overexpression data support the direct regulation of only *desA1/desA2,* among bicluster 276 genes. It is worth noting that EGRIN biclusters can be overlapping sets of co-regulated genes that, in some cases, group together genes from different regulons and, in other cases, subdivide genes of the same regulon, or even the same operon. This conditional modularity captures complex gene regulatory programs for combinatorial control for thousands of genes by few hundred TFs. While a significant number of bicluster 276 genes are up-regulated during *in vivo* infection, not all of the genes are necessarily regulated by the same TF. As such, bicluster 276 represents a coordination of regulatory mechanisms that bring together functionally related genes. These genes, involved in biosynthesis/transport of PDIM and desaturation of mycolic acids, act together to alter cell wall composition, thereby affecting cell wall permeability and host responses during *in vivo* infection.

### Inducible overexpression of *MSMEG_0916* or *Rv0472c* causes loss of mycobacterial viability and reduction in mycolate biosynthesis

Motivated by our identification of Rv0472c and *MSMEG_0916* as controlling the expression of *desA1* and *desA2*, we hypothesized that overexpression-mediated repression of the desaturases should have phenotypes similar to the *desA1* knockout that was previously characterized (Singh et al, 2016). We tested the viability of the TF overexpression strains by spotting serial 10-fold dilutions of cultures on agar plates with or without ATc. Plates with MTB and *Rv0472c* overexpression strain were incubated for 3 weeks, and growth patterns indicated that the presence of ATc resulted in a 4-log fold reduction in CFU counts (Fig 5A). In comparison, plates containing the parental MTB strain showed no change in CFUs with the presence or absence of ATc (Fig 5A). Similar experiments were done in *Mycobacterium bovis* BCG (BCG) and MSM with *Rv0472c* and *MSMEG_0916* overexpression, respectively. Overexpression resulted in 3-log viability reduction in BCG (Appendix Fig S6A) and 2-log viability reduction in MSM (Appendix Fig S6B). We also observed very limited growth in broth culture when *MSMEG_0916* was induced with ATc (Appendix Fig S7).

Overexpression of the conserved TF resulted in a loss of mycobacterial viability, due to repression of *desA1* and *desA2* and ensuing decrease in mycolic acid biosynthesis. Conditional depletion of DesA1 in MSM leads to an intermediate decrease in desaturation prior to complete loss of mycolic acids (Singh et al, 2016). To test for a decrease in mycolic acid biosynthesis, we labeled cultures of *MSMEG_0916* overexpression strain with $^{14}$C acetic acid following growth in the presence or absence of ATc. Thin layer chromatography (TLC) analysis of apolar lipids demonstrated that overexpression of *MSMEG_0916* reduced the levels of trehalose dimycolates (TDMs; Appendix Fig S8). We also analyzed methyl esters of mycolic acids (MAMEs) obtained from apolar lipids using 2D-argentation TLC analysis, designed to separate each subclass of mycolic acid based on saturation levels. MAME analysis revealed an accumulation of products that migrate identically to our previous observation with DesA1 depletion (Singh et al, 2016) (Fig 5B) and most likely correspond to mono-unsaturated mycolates. Similarly, 1D TLC separation of fatty acid methyl esters (FAMEs) and MAMEs from apolar lipids confirmed the general decrease in MAMEs and an accumulation of FAMEs when Rv0472c is overexpressed in BCG (Fig 5C and densitometric analysis in Appendix Fig S9). This characteristic profile of total MAME inhibition and FAME accumulation mirrors what is seen with fatty acid synthase (FAS)-II inhibitors, such as isoniazid (Vilcheze *et al*, 2000) and thiolactomycin (Kremer *et al*, 2000), and confirms the involvement of DesA1 and DesA2 in the biosynthesis of mycolic acids and more specifically with the FAS-II system. Interestingly, in BCG there was no accumulation of mono-unsaturated mycolates as those found in MSM upon detailed

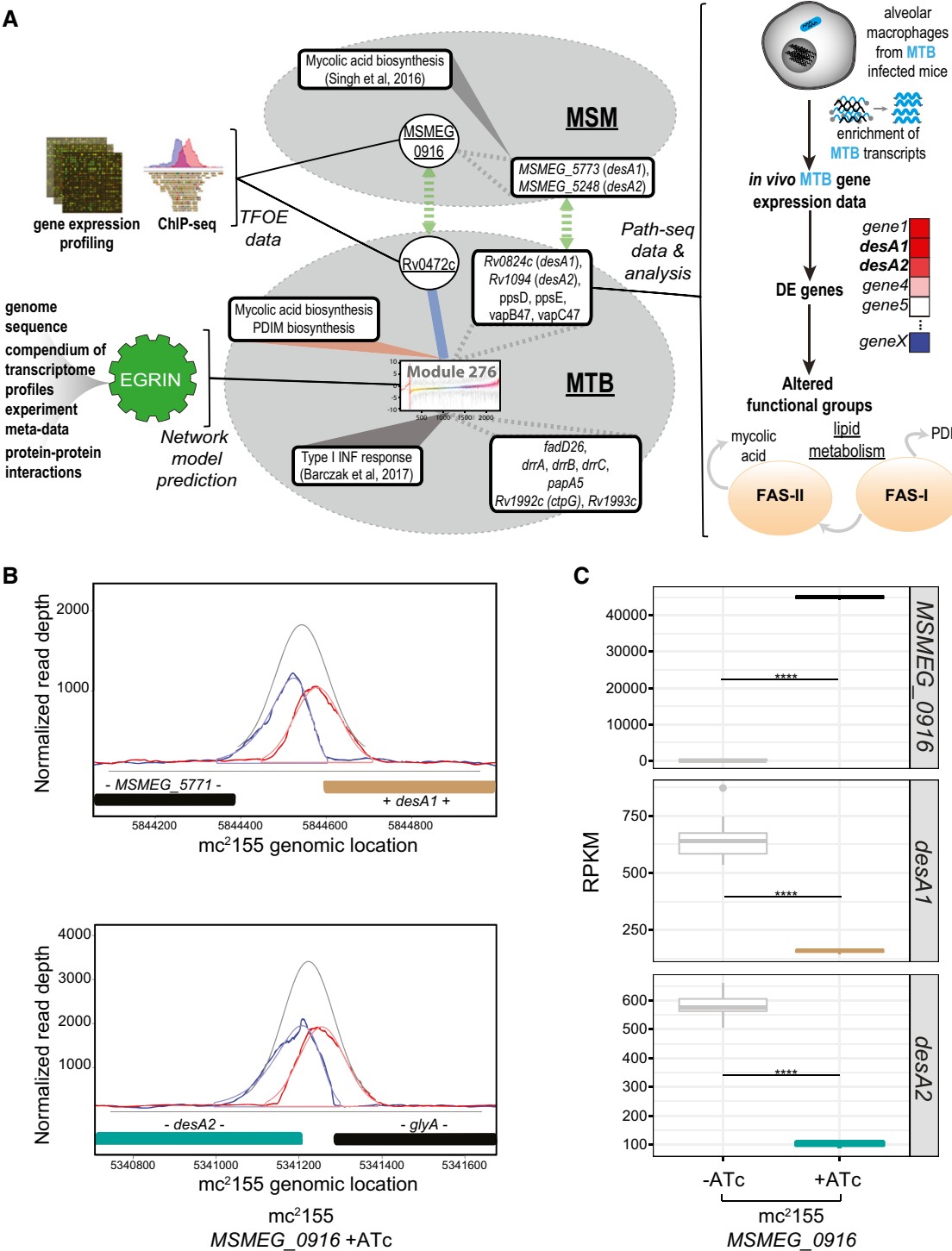

**Figure 4. EGRIN model predicts module 276 relevant to *in vivo* infection and regulation of *desA1* and *desA2* by Rv0472c (*MSMEG_0916*).**

A   EGRIN model of MTB predicts a module enriched with lipid metabolism genes from Path-seq data of *in vivo* infection. Overexpression data confirm the regulation of *desA1* and *desA2* by Rv0472c in both MTB and MSM. Graphic representation of linkages between module 276 genes, regulatory influences, functional associations, cell wall modifications, and homology to MSM.

B   Plot of read pile-ups from MSM with inducible overexpression of *MSMEG_0916* shows ChIP binding in the promoters of *desA1* and *desA2*.

C   Boxplots representing RPKM values from RNA-seq of MSM with inducible overexpression of *MSMEG_0916*. Significant log2 fold change (FC) between uninduced (−ATc) and induced (+ATc) samples for *MSMEG_0916* (log2 FC = 4.99), *desA1* (log2 FC = −1.32), and *desA2* (log2 FC = −1.8) with multiple hypothesis-adjusted *P*-values < 0.0001 as calculated with DuffyNGS (see Materials and Methods). Data are from three biological samples (induced) and nine biological samples (uninduced). The boxes show the median and first/third quartiles of the log2 FC values; the whiskers extend to the smallest/largest values that are no further than 1.5 times the inter-quartile range.

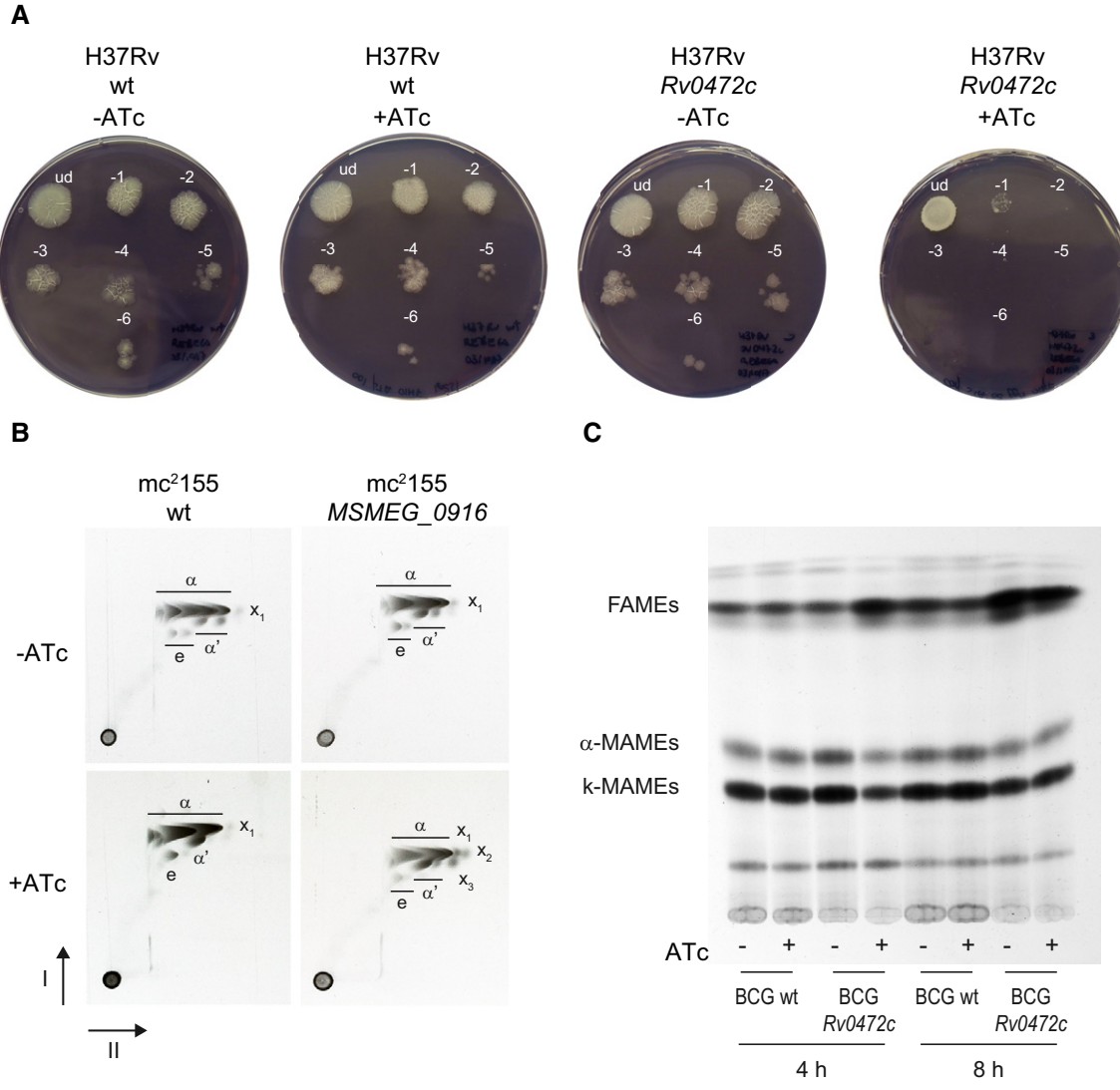

**Figure 5. Cell viability and mycolic acid characterization.**

A   Serial 10-fold dilutions of MTB H37Rv wild type (wt) and MTB with inducible overexpression of *Rv0472c* were spotted on 7H10 agar plates with or without ATc.

B   Argentation TLC of $^{14}$C-labeled methyl esters of mycolic acids (MAMEs) obtained from apolar lipids and delipidated cell wall fractions of MSM wt and MSM with inducible overexpression of *MSMEG*_0916. The $\alpha$, $\alpha'$, epoxy (e), and cyclopropanated $\alpha$- ($X_1$) MAMEs species are labeled. Faster-migrating species that co-migrated with $\alpha$-MAMEs and accumulate with induced *MSMEG*_0916 overexpression are indicated as $X_2$ and $X_3$.

C   BCG wt and BCG with inducible overexpression of *Rv0472c* cultures, labeled with $^{14}$C-acetate, were induced (+ATc) or uninduced (−ATc) for 4 h or 8 h. The total FAMEs and MAMEs were extracted and analyzed by autoradiography–TLC using equal counts (15,000 cpm) for each lane.

analysis of MAMEs by 2D-argentation TLC (Appendix Fig S10A and B). This could be due to key differences in mycolate subclasses between MSM and MTB, particularly cyclopropane ring formation, which is abundant in MTB but not MSM and requires a precursory desaturation event.

# Discussion

During MTB infection, the bacterium utilizes various mechanisms to ensure its own survival and persistence in the host. The intracellular context is paramount for identifying such mechanisms via observation and interpretation of gene expression changes. While hypoxia, low pH, and nutritional stress are used as proxies, they do not reproduce the spatiotemporal complexity of host-induced stress. As such, there is no substitute for understanding the authentic intracellular context other than transcriptionally profiling pathogens directly from infected cells and tissues. The development of Path-seq has enabled such studies and confirmed mycolic acid biosynthesis as a well-known virulence factor (Barry *et al*, 1998). Mycolates, essential for mycobacterial cell wall rigidity, not only make up a lipid-rich barrier in the mycobacterial cell envelope, they also act as potent immunomodulators, driving the pathogenesis of MTB, primarily as part of the cord factor (TDM; Marrakchi *et al*, 2014;

Nataraj *et al*, 2015). Here, we present evidence that mycolate biosynthesis is tightly regulated in response to the intracellular environment. Using our novel Path-seq method, we observed significantly induced expression of mycolic acid biosynthesis genes, *umaA*, *pcaA*, *desA1*, *desA2*, *fadD32*, and *fabD*, 24 h after MTB infection of mice. Using multiple systems-level approaches to understand the regulatory control of these virulence genes, we validated that *desA1* and *desA2* are regulated by Rv0472c (*MSMEG*_0916) and that Rv0472c-mediated repression leads to reduced mycolate biosynthesis and loss of mycobacterial viability. As DesA1 and DesA2 have been shown to be involved in mycolic acid biosynthesis via desaturation of the merochain, we have therefore named their transcriptional regulatory protein MadR (for mycolic acid desaturase regulator).

Not much is known about the regulation of mycolic acid biosynthesis apart from two transcription factors shown to regulate distinct operons, both containing genes encoding core FAS-II proteins (Salzman *et al*, 2010; Jamet *et al*, 2015). Studying the regulatory alterations to mycolate subclasses remains an even greater challenge, especially during infection. Our studies show that MadR is involved in the *in vivo* and *in vitro* regulation of *desA1* and *desA2*, coding for enzymes involved in mycolic acid desaturation (Singh et al, 2016). The introduction of double bonds in the mycolate merochain precedes cyclopropanation and other merochain modifications that are critical for pathogenic mycobacteria (Glickman *et al*, 2000; Rao *et al*, 2005, 2006; Barkan *et al*, 2012). Loss of cyclopropanation can lead to hyperinflammatory responses and attenuated infection. As the introduction of double bonds in the merochain is required for subsequent cyclopropanation and other merochain modifications, DesA1 and DesA2 could be drivers of both mycolic acid biosynthesis and composition during infection. In other words, MadR-driven regulation not only leads to lower mycolate levels during dormancy, a state when new cell wall material is not synthesized, but also altered cyclopropane ring formation by varying desaturation levels, thus affecting virulence and persistence.

Surprisingly, we observed early (2 h post-infection) induced expression of *desA1* and *desA2* during MTB infection of BMDMs, followed by return to basal levels by 8 h. This is consistent with the reported increased production of TDM within the first 30 min after *in vitro* phagocytosis (Fischer *et al*, 2001) and suggests that the desaturases play a role in cell wall modifications that occur in response to intracellular cues. However, the presence of these intracellular cues appears to be different in AMs from infected mice. The overall disparity in the transcriptional profile of MTB from BMDMs and AMs is both intriguing and disturbing. The active MTB networks we identified from BMDMs imply the presence of early and sustained bacterial stress. However, the induction of these stress-related networks is absent in the transcriptomes of MTB from AMs, suggesting the bacteria are not experiencing the same type or amount of stimuli in AMs. These data support recent observations using fluorescent MTB reporter strains, demonstrating that bacilli in AMs exhibit lower stress and higher bacterial replication than those in interstitial macrophages (Huang *et al*, 2018). Similarly, we

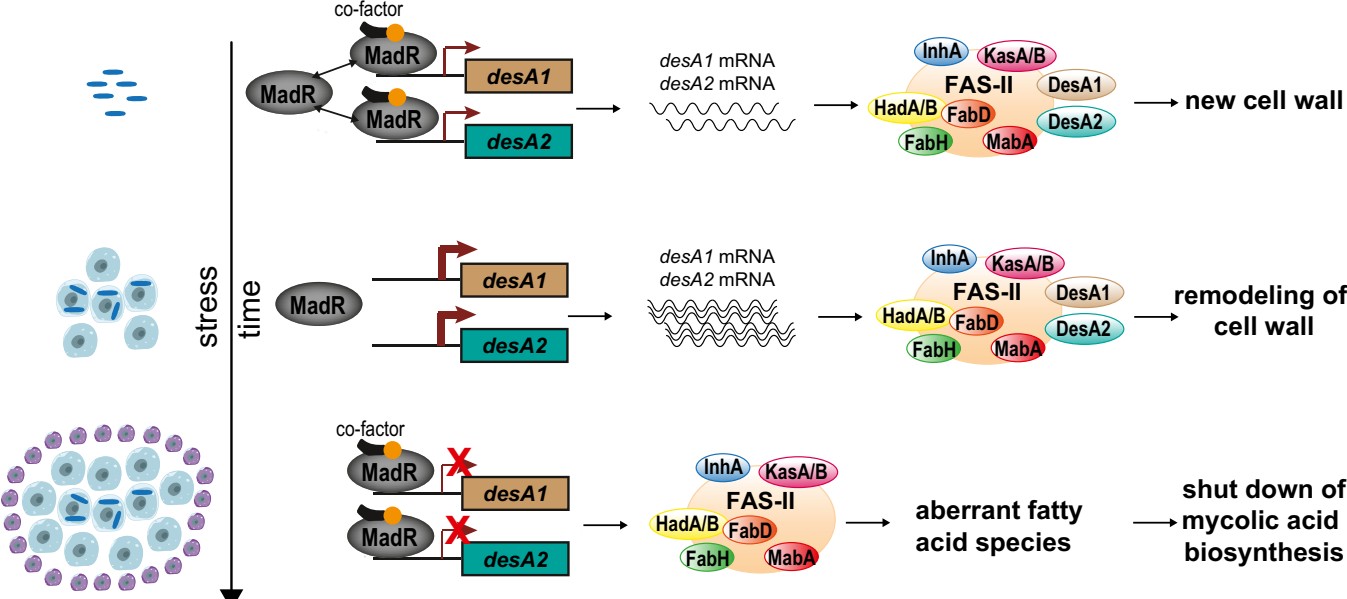

**Figure 6.  Model of MadR regulation of mycolic acid desaturases.**
Under normal growth conditions, MadR exists in equilibrium between the free and DNA-bound forms, and a basal level of *desA1* and *desA2* expression is maintained. DesA1 and DesA2 act with other enzymes of the fatty acid synthase-II (FAS-II) complex to produce mycolic acids for new cell wall. Infection of macrophages and early stress cues shifts the equilibrium toward the free form of MadR, and the repression of *desA1* and *desA2* is released. The desaturase protein levels increase, introducing double bonds that allow cyclopropanation and other modifications to alter the cell wall. These changes enable the bacteria to withstand intracellular stresses and establish infection. As infection continues and stress is sustained, MadR binds tightly to the promoter of desA1 and desA2, leading to stringent repression of the desaturases. MadR-mediated repression of desA1 and desA2 leads to irregular fatty acids of medium length and the pausing of mycolic acid biosynthesis to enter into dormancy. The DNA-bound form of MadR is shown in complex with a yet unknown co-factor that leads to repression of the desaturases.

hypothesize that MTB responds divergently to macrophages of different lineages and that AMs present fewer stresses and possibly a more permissive environment compared to BMDMs. It is also worth mentioning that the data raise some concerns with respect to the use of BMDMs as an appropriate infection model.

Our data lead us to propose a model for MadR regulation of *desA1* and *desA2* transcription as summarized in Fig 6. Under normal growth conditions, MadR exists in equilibrium between the free and DNA-bound forms, thus maintaining basal levels of *desA1* and *desA2* transcripts. Upon macrophage infection and early hypoxia, equilibrium favors unbound MadR which derepresses *desA1* and *desA2* transcription and increases mRNA levels. As infection progresses and reaches later stages of hypoxia, MadR has increased binding affinity in the promoters of *desA1* and *desA2* and represses their transcription to below basal levels. Ultimately, the MadR regulatory system enables mycobacteria to efficiently alter mycolate biosynthesis and composition in response to environmental signals. We suspect the early response to infection (*desA1* and *desA2* up-regulation) increases desaturation events and allows MTB to fine-tune cyclopropanation and other merochain modifications that contribute to the establishment of infection. However, mycolate biosynthesis is energetically expensive and MadR-mediated repression occurs in later stages of infection. The reduction in mycolate biosynthesis allows MTB to enter dormancy and facilitates long-term persistence.

The question remains how MadR is able to differentially bind to DNA in response to environmental changes. In mycobacteria, other TFs regulating mycolate biosynthesis are modulated by long-chain acyl-CoAs (Biswas *et al*, 2013; Mondino *et al*, 2013; Tsai *et al*, 2017), proposing a role for these molecules in the modulation of MadR as well. Similarly, a MadR homolog in *Pseudomonas aeruginosa*, DesT, was shown to have enhanced DNA binding in the presence of unsaturated acyl-CoAs (Zhang *et al*, 2007). These studies support the notion that a select acyl-CoA ligand may control MadR DNA binding affinity (as shown in Fig 6) and thus the expression of *desA1* and *desA2*.

The characterization of the MadR regulon provides valuable insight for understanding the evolution of MTB. While we have shown the regulation by MadR is conserved from MSM to MTB, our results also suggest the fatty acid desaturation events and resulting mycolate subclasses and surface lipids have evolved, specializing for bacterial survival in the host environment. These findings propose mycobacterial evolution from saprophyte to pathogen has occurred through the adaptation of ancestral genes and regulatory networks to function in the host environment. Ultimately, this study demonstrates the *in vivo* significance of the desaturases and their regulation by MadR. We believe the Path-seq method, described and employed here, offers a sensitive and tractable approach to elucidate the molecular mechanisms used by MTB during host infection, potentially at the single-cell level. Our detailed characterization of one such mechanism has revealed that modulation of MadR activity can affect cell wall composition as well as mycobacterial viability. Accordingly, we have established Path-seq as a powerful tool for uncovering the minimally studied *in vivo* biology of this pathogen and revealed the essentiality of MadR encoded program for cell wall remodeling and biosynthesis. As such, we present MadR as a new and important antitubercular target.

# Materials and Methods

## Reagents and Tools table

| Reagent/Resource | Reference or Source | Identifier or Catalog Number |
|---|---|---|
| **Experimental Models** | | |
| H37Rv (*M. tuberculosis*) | D. Sherman lab | N/A |
| mc²155 (*M. smegmatis*) | A. Bhatt lab | N/A |
| BCG Pasteur (*M. bovis*) | A. Bhatt lab | N/A |
| C57BL/6J (*M. musculus*) | Jackson Lab | B6.129P2Gpr37tm1Dgen/J |
| **Recombinant DNA** | | |
| pDTCF-Msmeg0916 (*M. smegmatis*) | This study | N/A |
| pDTCF-Rv0472c (*M. tuberculosis*) | D. Sherman lab, Minch *et al* (Data ref: Minch *et al*, 2015) | N/A |
| pMV306-eGFP-Zeo | L. Kremer lab, Bernut *et al* (2016) | N/A |
| **Antibodies** | | |
| M2 anti-FLAG | Sigma | F1804 |
| Rat anti-mouse CD16/32 (clone 2.4G2) | BD Pharmingen | 553142 |
| Mouse anti-Siglec F (clone E50-2440) | BD Pharmingen | 552125 |
| Mouse anti-CD11b (clone M1/70) | BioLegend | 101201 |
| Mouse anti-CD64 (clone X54-5/7.1) | BioLegend | 139321 |
| Mouse anti-CD45.2 (clone 104) | BioLegend | 109815 |
| Mouse anti-CD3 (clone 17A2) | eBioscience | 14-0032-81 |

**Reagents and Tools table**   (continued)

| Reagent/Resource | Reference or Source | Identifier or Catalog Number |
|---|---|---|
| Mouse anti-CD19 (clone 1D3) | eBioscience | 11-0193-81 |
| **Oligonucleotides and sequence-based reagents** | | |
| Cloning primers | This study | see Methods and Protocols |
| **Chemicals, enzymes and other reagents** | | |
| Zombie Violet dye | BioLegend | 423113 |
| TRIzol | Invitrogen | 15596026 |
| RPMI Medium (1640) | Gibco | 11875093 |
| Anhydrotetracyline | Sigma-Aldrich | 37919 |
| Hygromycin B | Invitrogen | 10687010 |
| Protease inhibitor cocktail | Sigma-Aldrich | P2714 |
| Middlebrook OADC enrichment | BD Difco | 212351 |
| Middlebrook ADC enrichment | BD Difco | 212352 |
| Middlebrook 7H10 agar | BD Difco | 262710 |
| Middlebrook 7H9 broth | BD Difco | 271310 |
| Petroleum Ether 60–80°C | Fisher Chemicals | P/1480/17 |
| Chloroform, 99.8+% | Fisher Chemicals | C/4960/17 |
| Methanol, AR | Fisher Chemicals | M/4000/17 |
| Sodium chloride | Fisher Chemicals | S/3160/65 |
| Tetrabutylammonium hydroxide solution 40 wt. % in $H_2O$ | Sigma-Aldrich | 178780 |
| Dichloromethane, 99+% | Fisher Chemicals | D/1850/17 |
| Iodomethane (stabilised with silver) for synthesis | Merck | 806064 |
| Diethyl ether, AR | Fisher Chemicals | D/2450/17 |
| TLC Silicagel 60 $F_{254}$ | Merck | 1055540001 |
| Acetone, AR | Fisher Chemicals | A/0600/17 |
| Silver nitrate, Ultrapure Grade, 99.5% | Acros Organics | 419361000 |
| Acetic Acid, Sodium Salt, [1-14C]-, 1mCi (37MBq) | PerkinElmer | NEC084H001MC |
| Kodak® BioMax® MR film | Sigma-Aldrich | Z353949-50EA |
| **Software** | | |
| DuffyNGS | Vignali *et al* (2011) http://networks.systemsbiology.net/mtb/ | |
| NetSurgeon | Michael *et al* (2016) http://mblab.wustl.edu/software.html | |
| DESeq2 | Love *et al* (2014) https://github.com/mikelove/DESeq2 | |
| Adobe Photoshop CC 2015 | https://www.adobe.com | |
| **Other** | | |
| Direct-zol RNA MicroPrep kit | Zymol Research | R2060 |
| Dynabeads M-270 Streptavidin | Invitrogen | 65306 |
| Lysing Matrix B tubes | MP Biomedicals | MP116911050 |
| SureSelect$^{XT}$ strand-specific RNA kit | Agilent | 5190-6410 |
| SureSelect$^{XT}$ target enrichment kit | Agilent | 5190-4393 |
| Probe library M_tub_h37rv_ASM19595v2_32_1 | Agilent | This study, ELID number 3037441 |
| TruSeq stranded mRNA library prep kit | Illumina | RS-122-2103 |
| Ribo-Zero magentic kit bacteria | Illumina | MRZB12424 |
| Ribo-Zero magentic kit epidemiology | Illumina | MRZE724 |
| NextSeq 500/550 High output kit | Illumina | FC-404-2002 |
| SMARTer ThruPLEX DNA-seq kit | Takara | R400523 |
| Illumina NextSeq 500 | Illumina | |

## Methods and Protocols

### Culturing conditions

Mycobacteria strains were cultured in Middlebrook 7H9 with the ADC supplement (Difco), 0.05% Tween-80 at 37°C under aerobic conditions with constant agitation. Strains containing the anhydrotetracycline (ATc)-inducible expression vector were grown with the addition of 50 μg/ml hygromycin B to maintain the plasmid. Growth was monitored by OD600 and colony-forming units (CFUs). For experiments featuring *madR* overexpression strains, overexpression was induced for the approximate duration of one cell doubling (18 h for MTB and BCG, 4 h for MSM) using an ATc concentration of 100 ng/ml culture. Wild-type and overexpression strain cultures were grown into mid-log phase. For assessing growth on agar plates, broth cultures were adjusted to OD600 of 0.5, and serial dilutions were spotted on 7H10 containing 0.5% (v/v) glycerol and 10% (v/v) OADC plates, with or without 100 ng/ml ATc. In the case of the overexpression strain, 50 ng/ml hygromycin was added to the solid medium. For growth in broth, MSM mid-logarithmic phase cultures containing the integrative vector pMV306-eGFP-Zeo (Bernut *et al*, 2016) were inoculated in an initial OD600 0.05 in 200 μl of 7H9 supplement with 0.2% (v/v) glycerol, 0.05% (v/v) Tween-80, and 10% (v/v) OADC with or without 100 ng/ml ATc in sealed 96-well Costar 3603 black-sided clear-bottomed plate incubated at 37°C. Fluorescence was acquired every 30 min for 30 h at 37°C, in a PHERAstar FS microtiter plate reader (BMG Labtech), using 485 nm and 520 nm as excitation and emission wavelengths, respectively. When growing BCG or MSM for lipid extraction, cultures were cultured up to OD600 0.5. Then, they were induced with 100 ng/ml ATc (Sigma-Aldrich) final concentration and labeled with acetic acid [1–14C] 1 mCi/ml (PerkinElmer), if hot lipid analysis was going to be performed. MSM samples were collected the following day, while BCG cultures at 4 and 8 hours post-induction.

### Strains

To investigate the growth properties of MadR overexpression, we used strains containing an ATc- inducible expression vector of the gene, as described previously (Galagan *et al*, 2013; Jaini *et al*, 2014; Rustad *et al*, 2014; Data ref: Minch *et al*, 2015). The pDTCF-*Rv0472c* (kind gift of David Sherman) was transformed into *M. bovis* BCG and *M. tuberculosis* H37Rv for viability and mycolic acid characterization. *Mycobacterium tuberculosis* H37Rv (kind gift of David Sherman) was also used for Path-seq experiments. For MadR overexpression in *M. smegmatis* (*MSMEG_0916*), we created entry clones through PCR amplification of the gene template from mc²155 gDNA, adding the necessary Gateway recombination sequences to the PCR product, as described in Minch *et al* (Data ref: Minch *et al*, 2015). The primers used for the Gateway entry cloning pDONR221 vector are 5′-GGGGACAAGTTTGTACAAAA AAGCAGGCTCTGTGGCACAGCAGACTCCACCG-3′ (forward) and 5′-GGGGACCACTTTGTACAAGAAAGCTGGGTCGTCAAGCAGGTGC CGCGGCGG-3′ (reverse). We inserted the gene into the same *E. coli*-mycobacterial episomal shuttle vector (pDTCF) modified as described in Minch *et al* (Data ref: Minch *et al* 2015). The pDTCF-*MSMEG0916* plasmid was transformed into *M. smegmatis* mc²155.

### Mice

C57BL/6 mice were purchased from the Jackson Laboratory. All mice were housed and bred under specific pathogen-free conditions at Seattle Children's Research Institute (SCRI). All experimental protocols involving animals were approved by the Institutional Animal Care and Use Committee of (SCRI).

### Aerosol infection

A mid-log-phase stock of MTB H37Rv was used to infect mice in an aerosol infection chamber (Glas-Col). Bacterial load in the lungs was determined by plating serial dilutions from homogenized lungs.

### Cell isolation, analysis, and sorting

1  Bronchoalveolar lavage was performed by first exposing the trachea of euthanized mice.
2  The exposed trachea was punctured using Vannas Micro Scissors (VWR), and 1 ml PBS was injected using a 20G-1" IV catheter (McKesson) connected to a 1-ml syringe.
3  The PBS was flushed into the lung and aspirated three times, and the recovered fluid was placed in a 15-ml tube on ice.
4  The 1 ml PBS wash was then repeated three additional times for a total of 4 ml recovered fluid.
5  Cells were filtered, spun down, and resuspended in a 96-well plate for antibody staining.
6  Cells were suspended in 1X PBS (pH 7.4) containing 0.01% $NaN_3$ and 1% fetal bovine serum (i.e., FACS buffer).

Fc receptors were blocked with anti-CD16/32 (2.4G2, BD Pharmingen). Cell viability was assessed using Zombie Violet dye (BioLegend). Surface staining included antibodies specific for murine Siglec F (E50-2440, BD Pharmingen), CD11b (M1/70, BioLegend), CD64 (X54-5/7.1, BioLegend), CD45 (104, BioLegend), CD3 (17A2, eBiosciences), and CD19 (1D3, eBiosciences). Cell sorting was performed on a FACSAria (BD Biosciences). Cells were collected in complete media, spun down, resuspended in TRIzol, and frozen at −80° overnight prior to RNA isolation.

### BMDM infection

BMDMs were isolated from C57BL/6J mice and cultured in RPMI (RPMI containing 10% (v/v) FBS and 2 mM L-glutamine) with recombinant human CSF-1 (50 ng/ml) for 6 days and then replated. BMDMs were infected on Day 7 with MTB H37Rv strain (MOI 10), followed by washing 3× with RPMI at 2 h post-infection, and fresh media was added. BMDMs were lysed with TRIzol (Invitrogen), and total RNA was isolated from mixed host–pathogen sample. The same MTB H37Rv cultures used for BMDM infection were also diluted to starting $OD_{600}$ = 0.1 and grown in 7H9.

### Hypoxia time course

An Oxygen Sensor Spot (PreSens, Regensburg, Germany) was adhered within a 1-l disposable spinner flask with two side arms (Corning, Corning, NY). A velcro belt with a screw-on port for the fiber optic cable was wrapped around the flask. A gas line input was fastened on one arm of the flask, and a Luer-Lock/ filter sampling port was connected to the other arm. Air and $N_2$ gas lines were run into the Biological Safety Laboratory and connected to gas-specific mass flow controllers (Alicat, Tucson, AZ),

whose outputs were connected downstream through a Y-connector that led into an incubator. Three separate flasks, all prepared as described above, were placed onto a stir plate inside an incubator at 37°C. The mixed gas line was split via additional Y-connectors, streamed through 0.2-μm filters, and attached to the gas line inputs of each flask. Media was incubated overnight and checked for contamination before inoculated with MTB.

The mass flow controllers and oxygen sensor were linked to a computer and remotely controlled and monitored in real time. After inoculation of 700 ml 7H9 media with MTB H37Rv at starting OD600 of 0.01, we programmed the mass flow controllers to achieve a changing gas mixture gradient, which allowed us creating a steady 2-day depletion, followed by 2 days of sustained hypoxia, and reaeration by flowing pure air into the headspace of the vessels and increasing the speed of the stir bars in each vessel. Samples were collected by attaching a Luer-Lock syringe to the sampling port. Samples were centrifuged at high speed for 5 min, supernatant was discarded, and cell pellet was immediately flash frozen in liquid nitrogen.

### RNA isolation

Cell pellets in TRIzol were transferred to a tube containing Lysing Matrix B (MP Biomedicals) and vigorously shaken at max speed for 30 s in a FastPrep 120 Homogenizer (QBiogene) three times. This mixture was centrifuged at max speed for 1 min, and the supernatant was transferred to a fresh tube. RNA from extracellular MTB samples, BMDM infection, hypoxia time course, and madR overexpression was isolated using the Direct-zol RNA MicroPrep Kit (Zymol Research) according to manufacturer's instruction with on-column DNase treatment.

RNA from mice infection was isolated by the following method:
- 200 μl chloroform was added to 1 ml of TRIzol.
- Samples were inverted and incubated for 2–3 min, and the upper aqueous phase was collected.
- A second chloroform extraction was done, followed by addition of 1 μl glycogen and 500 μl isopropanol.
- Samples were incubated with isopropanol for 10 m at room temperature and centrifuged, and supernatant was discarded.
- Pellet was washed with 1 ml 70% ethanol twice.
- All ethanol was removed, the pellet dried (15 m), and resuspended in 12 μl RNase-free water.

For all RNA samples, total RNA yield was quantified by NanoDrop (Thermo Scientific) and quality was analyzed in a 2100 Bioanalyzer system (Agilent Technologies). Following DNase treatment, total RNA samples were depleted of ribosomal RNA using the Ribo-Zero Gold rRNA Removal Kit "epidemiology" (Illumina).

### Probe design

Non-overlapping head-to-tail 120-nucleotide probes were designed using the Array software (Agilent Technologies). A total of 35,624 probes were designed to cover 3,924 *M. tuberculosis* H37Rv ORFs (Agilent probe library M_tub_h37rv_ASM19595v2_32_1, ELID number 3037441). Using Megablast, it was verified that all genes of MTB were matched by at least one probe and that only a negligible fraction of the probes could be mapped on the mouse and human cDNA sequences from Ensembl.

### Preparation of libraries for transcriptional sequencing

RNA libraries for Path-seq were prepared using the SureSelect[XT] strand-specific RNA target enrichment for Illumina multiplexed sequencing. RNA libraries for RNA-seq were prepared using the SureSelect[XT] strand-specific RNA kit, but were not hybridized to probes and indexed separately.

Briefly, the protocol followed was as follows:
- RNA from rRNA-depleted samples was enzymatically fragmented, and double-stranded cDNA was produced with adapters ligated to both ends.
- The library was then amplified using provided primers which hybridize to the previously inserted adapters, therefore allowing a linear amplification to all transcripts present in the sample. In the case of non-enriched RNA-seq samples, sample indexes were also inserted during this PCR.
- For Path-seq libraries, double-stranded cDNA ligated to adapters was also amplified and then incubated at 65°C for 24 h with the set of biotinylated oligonucleotides specifically designed to capture MTB transcripts, as described above.
- The hybridized sequences were captured with magnetic streptavidin beads (M-270, Invitrogen).
- They were next linearly amplified using provided primers and indexed during PCR.

Before sequencing, libraries were assessed for quality and fragment size by Bioanalyzer and with a Qubit Fluorometer (Invitrogen) to determine cDNA concentration. Resulting libraries were sequenced on the Illumina NextSeq Instrument using mid output 150 v2 reagents. Paired-end 75 bp reads were processed following Illumina default quality filtering steps.

### Transcription abundance from sequencing data

Raw FASTQ read data were processed using the R package DuffyNGS as described previously (Vignali et al, 2011). Briefly, raw reads pass through a 3-stage alignment pipeline: (i) a prealignment stage to filter out unwanted transcripts, such as rRNA, mitochondrial RNA, albumin, and globin; (ii) a main genomic alignment stage against the genome(s) of interest; and (iii) a splice junction alignment stage against an index of standard and alternative exon splice junctions. Reads from samples of mixed host–pathogen RNA and extracellular MTB controls were aligned to a combined *M. tuberculosis H37Rv* (ASM19595v2) and *Mus Musculus* (GRCm38.p6) genome. Reads from samples of MSM RNA were aligned to *M. smegmatis* mc²155 genome (ASM1500v1). All alignments were performed with Bowtie 2 (Langmead & Salzberg, 2012), using the command line option "very-sensitive". BAM files from stages 2 and 3 are combined into read depth wiggle tracks that record both uniquely mapped and multiply mapped reads to each of the forward and reverse strands of the genome(s) at single-nucleotide resolution. Multiply mapped reads are prorated over all highest-quality aligned locations. Gene transcript abundance is then measured by summing total reads landing inside annotated gene boundaries, expressed as both RPKM and raw read counts. Two stringencies of gene abundance are provided using all aligned reads and by just counting uniquely aligned reads.

### Differential expression

For both infection models (*in vitro* and *in vivo*), we used DESeq2 (Love *et al*, 2014) to identify gene expression changes between

intracellular and extracellular MTB at each sampled time point. We used rounded raw read counts estimated by DuffyNGS (as described above) as input for DESeq2. Genes with absolute log2 fold change bigger than one and multiple hypothesis-adjusted *P*-value below 0.01 and 0.05, for the *in vitro* and *in vivo* data, respectively, were considered differentially expressed. For the *in vivo* samples, only genes with non-zero counts in any of the replicates were considered in DESeq2.

For *Msmeg0916* overexpression, we used a panel of 5 DE tools to identify gene expression changes between induced (+ATc) and uninduced (−ATc). The tools included (i) RoundRobin (in-house); (ii) RankProduct (Breitling *et al*, 2004); (iii) significance analysis of microarrays (SAM) (Tusher *et al*, 2001); (iv) EdgeR (Robinson & Smyth, 2008); and (v) DESeq2 (Love *et al*, 2014). Each DE tool was called with appropriate default parameters and operated on the same set of transcription results, using RPKM abundance units for RoundRobin, RankProduct, and SAM and raw read count abundance units for DESeq2 and EdgeR. All 5 DE results were then synthesized, by combining gene DE rank positions across all 5 DE tools. Specifically, a gene's rank position in all 5 results was averaged, using a generalized mean to the 1/2 power, to yield the gene's final net rank position. Each DE tool's explicit measurements of differential expression (fold change) and significance (*P*-value) were similarly combined via appropriate averaging (arithmetic and geometric mean, respectively). Genes with averaged absolute log2 fold change bigger than one and multiple hypothesis-adjusted *P*-value below 0.01 were considered differentially expressed.

### MTB signed transcriptional network

We compiled a signed (stating the positive or negative nature of each TF–gene interaction) wiring diagram of MTB transcriptional regulatory network. The compiled MTB network included 4,635 TF–gene interactions (2,296 and 2,339 instances of activation and repression, respectively) with both physical (detected with ChIP-seq experiments) and functional evidence (detected with transcriptional profiling). The compiled network contained 2,001 genes and 136 TFs with at least one target. The initial ChIP-seq derived MTB network consisted of 6,581 interactions occurring in the −150 bp to +70 bp region of genes' promoter reported by Minch *et al* (Data ref: Minch *et al* 2015). We expanded that MTB ChIP-seq network by taking into account operon organizations. For a given TF–gene interaction, if the target gene is part of an operon, we included all other members of the operon as potential targets of the corresponding TF. The expanded MTB ChIP-seq network contained 12,188 interactions. Finally, we filtered out interactions that did not change at least 20% in the relevant TF-overexpressing strain (compared to the WT strain). Up-regulation of the target gene in the TF-overexpressing strain was interpreted as positive interaction (the opposite for down-regulation).

### Identification of transcription factors with differential activity in intracellular MTB (using NetSurgeon)

We identified potential TFs with increased or decreased regulatory activity in intracellular MTB (respect to extracellular controls) at each sampled time point using the method recently developed by Michael *et al* (2016) called the NetSurgeon algorithm. Briefly, NetSurgeon identifies TFs whose differential regulatory activity is likely responsible for the observed transcriptional changes between

two states of interest. In our case, we wanted to identify TFs that drive differential expression between intracellular MTB and their controls. Changes in TF activities are estimated based on the expression of their target genes (derived from DESeq2 output). TF regulons are extracted from a signed transcriptional regulatory network specified by the user. The signed MTB transcriptional network model used in this study is described above (and available at: http://net works.systemsbiology.net/mtb). NetSurgeon's scoring is based on the hypergeometric test distribution (Michael *et al*, 2016). Three important NetSurgeon's considerations are as follows: (i) Increase and decrease in TF activity are independently scored; (ii) only target genes differentially expressed (according to user's defined *P*-values, *q*-values, and fold change cutoffs) in the proper direction impact TF scores. This means that in case of increased activity, only genes significantly down-regulated and up-regulated will contribute to the score of their repressors and activators, respectively; and (iii) TF scores are defined not only by the number of target genes that are differentially expressed in the correct direction, but also by their adjusted *P*-values (associated with the differential expression analysis performed with DESeq2). The weight of each gene in the scores of its transcriptional regulators is proportional to the –log2 of its adjusted *P*-value. Thus, NetSurgeon's score is not homogeneous distributed among TFs' targets but it is affected by the differential expression significance of each gene. To reduce false positives (i.e., misleading presence of TFs with small number of known targets at the top of our TF scores ranking) due to overlap between regulons, only TFs with at least five targets were considered in this analysis. Furthermore, we considered TFs in the top 15 of NetSurgeon's score ranking as the ones with differential activity. This threshold was selected based on the NetSurgeon's results when we compared susceptible and antibiotic resistance *Escherichia coli* transcriptomes. Among the top 15 TFs, we were able to recall TFs known to be involved in multidrug resistance (such as MarA and homolog Rob, Ruiz & Levy, 2010). TFs with < 5 targets with differential expression or with contradicting trends (similar number of activated and repressed targets moving in the same direction) were excluded from the NetSurgeon's top 15 ranking. For the *in vivo* and 8 h *in vitro* data, the next two TFs outside the top 15 ranking were included to compensate for multiple TFs that were filtered out.

### ChIP-seq

The *madR* overexpression strain was induced for the approximate duration of one cell doubling (4 h for MSM) using an ATc concentration of 100 ng/ml culture. DNA–protein interactions were characterized as described previously (Data ref: Minch *et al*, 2015). Libraries were prepared using ThruPLEX DNA-seq Kit (Takara) using standard protocol. Samples were sequenced on the Illumina 550 NextSeq instrument, generating unpaired 20–30 million 75-bp reads per sample. Raw FASTQ read data were processed using the R package DuffyNGS as described previously (Vignali et al, 2011). For consensus motif determination, we searched for conserved DNA sequences within ± 50 nucleotides of high-quality (score > 0.7) ChIP-seq peak centers using MEME (Bailey & Elkan, 1994).

### Measuring viability of madR overexpression strains

Wild-type and *madR* overexpression strain cultures were grown into mid-log phase. For assessing growth on agar plates, OD of the broth culture was adjusted up to 0.5, and serial dilutions were spotted in

7H10 containing 0.5% (v/v) glycerol and 10% (v/v) OADC plates, with or without 100 ng/ml ATc. In the case of the overexpression strain, 50 ng/ml hygromycin was added to the solid medium.

### Extraction and analysis of total lipids and mycolic acids

Lipids were extracted from BCG and MSM cells in three fractions as describes by Dobson *et al* (1985), with a few modifications. Briefly, outside apolar lipids from dried pellets were extracted with two consecutive extractions with 4 ml of petroleum ether (60–80°C) and dried. Then, inside apolar and polar lipids were extracted following Dobson protocol.

Outside, inside apolar, and polar lipid extracts, along with delipidated pellets from MSM and BCG, were subjected to alkaline hydrolysis using tetrabutylammonium hydroxide (TBAH) as previously described (Kremer *et al*, 2002). Aliquots (15,000 cpm) from each outside, inside apolar, and polar lipid extracts were analyzed by thin layer chromatography (TLC) utilizing Silica Gel 60 F254 plates (Merck) developed once in the solvent system $CHCl_3/CH_3OH/H_2O$ (60:16:2, v/v/v). However, FAME and MAME aliquots (15,000 cpm) were resolved through TLC using petroleum ether/acetone (95:5, v/v) or by two-dimensional silver ion argentation thin layer chromatography (2D-TLC; Kremer *et al*, 2002). Autoradiograms were produced after exposing Carestream® Kodak® BioMax® MR film for 3 days. To determine the intensity of TLC spots, densitometric analysis using Adobe Photoshop CC 2015 was performed.

## Data availability

The datasets and computer code produced in this study are available in the following databases:

- *In vivo* Path-seq data: Gene Expression Omnibus GSE116394 https://www.ncbi.nlm.nih.gov/geo/query/acc.cgi?acc = GSE116394
- *In vitro* Path-seq/RNA-seq data: Gene Expression Omnibus GSE116357 https://www.ncbi.nlm.nih.gov/geo/query/acc.cgi?acc = GSE116357
- MadR RNA-seq data: Gene Expression Omnibus GSE116027 https://www.ncbi.nlm.nih.gov/geo/query/acc.cgi?acc = GSE116027
- MadR ChIP-seq data: Gene Expression Omnibus GSE116084 https://www.ncbi.nlm.nih.gov/geo/query/acc.cgi?acc = GSE116084
- Hypoxia/reaeration RNA-seq data: Gene Expression Omnibus GSE116353 https://www.ncbi.nlm.nih.gov/geo/query/acc.cgi?acc = GSE116353
- The Gene Regulatory Network to implement NetSurgeon: MTB Network Portal Data Center http://networks.systemsbiology.net/mtb/data-center
- R notebook with scripts for performing computation analyses: GitHub https://github.com/baliga-lab

**Expanded View** for this article is available online.

## Acknowledgements

We thank members of the Baliga, Bhatt, and Aderem laboratories for critical discussions; Albel Singh for help with graphics; and David Sherman and laboratory for providing the overexpression vectors and plasmids. Funding was provided by the National Science Foundation (1518261); Biotechnology and Biological Sciences Research Council (BB/N01314X/1); National Institute of Allergy and Infectious Diseases of the National Institutes of Health (R01AI128215, U19AI10676, U19AI135976); and MIBTP PhD Studentship to CC.

## Author contributions

EJRP, RB, ACR, AA, AB, and NSB designed research; EJRP, RB, ACR, AK, MP, DM, AAA, and CC performed research; EJRP and MLA-O performed computational analyses; EJRP, RB, ACR, MLA-O, AB, and NSB analyzed and interpreted data; and EJRP, RB, MLA-O, AB, and NSB wrote the paper.

## Conflict of interest

The authors declare that they have no conflict of interest.

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
