## [Review Process File · Molecular Systems Biology]

Path-seq identifies an essential mycolate remodeling program for mycobacterial host adaptation

Eliza J.R. Peterson, Rebeca Bailo, Alissa C. Rothchild, Mario L. Arrieta-Ortiz, Amardeep Kaur, Min Pan, Dat Mai, Abrar A. Abidi, Charlotte Cooper, Alan Aderem, Apoorva Bhatt and Nitin S. Baliga

Review timeline:

Submission date:	2 nd August 2018
Editorial Decision:	17 th September 2018
Revision received:	25 th January 2019
Editorial Decision:	30 th January 2019
Revision received:	31 st January 2019
Accepted:	4 th February 2019

Editor: Maria Polychronidou

Transaction Report:

1st Editorial Decision

17th September 2018

Thank you again for submitting your work to Molecular Systems Biology. We have now heard back from the three referees who agreed to evaluate your study. As you will see below, the reviewers think that the presented methodology and findings seem interesting. They raise however a series of concerns, which we would ask you to address in a major revision.

Overall, I think that the recommendations of the reviewers are rather clear and therefore there is no need to repeat the points listed below. Please feel free to contact me in case you would like to discuss in further detail any of the issues raised by the reviewers. Related to the presentation of the data, as reviewers #1 and #3 point out, the findings related to desA1 and desA2 need to be better incorporated in the narrative of the manuscript.

REFeree REPORTS

Reviewer #1:

Summary

This manuscript describes an exciting new technique (Path-seq) for transcriptomics which uses biotinylated oligos as baits to select *Mycobacterium tuberculosis* (Mtb) RNA. This has utility not just in monitoring dual host and pathogen transcriptomics from the same samples (the authors don't do this though) but also this library of probes has utility in single cell transcriptomics which is currently technically challenging on prokaryotes (the authors could point this out?). The authors describe validating this method using spiked samples of RNA from host and Mtb. They show that Path-seq enriches RNA from primary alveolar macrophages (AM) isolated from BAL samples 1000

fold. They go on to use this method to measure the transcriptomic profile of Mtb growing in AM and also murine bone marrow derived macrophages (BMDM) and compare with the in vitro transcriptomic profile. As the power of this method is to be able to in parallel assess the transcriptomic profile of the host and it's a shame that they didn't do this. Whilst the method is brilliant the key findings from this work were slightly disappointing as presented in the paper but I think this is perhaps due to a limited amount of data mining (discussed below). They find that the transcriptomic profile of the BMDM is highly similar to previous publications (further validating methodology) and rather surprisingly to the in vitro profile (growth rate effects, see below) whereas the in vivo profile was unique (although some genes were shared). Using a regulatory network the authors identify transcription factors with activity in their different systems. From here the authors seem to change gear from a global view to investigating two genes, *desA1* and *desA2* which are highly upregulated in vivo and early on in the ex vivo experiments but were not however in the identified regulons. Using EGRIN model they go on to identify the regulators controlling fatty acid desaturases using the in vivo data and identify Rv0472c. They then go on to perform chip-seq on *Mycobacterium smegmatis* (MS) to confirm that the MS version of Rv0472c regulated the expression of these genes (at least in this strain) and that overexpression of the regulator induced a similar phenotype to their previously published study.

Key findings

Novel methodology (Path-seq)

Transcriptional profile of Mtb growing in alveolar macrophages is very different (very few TF required) from primary bone marrow derived macrophages (behaving like Mtb in vitro)

DesA1 and *DesA2* are up regulated

Identification of the regulator of *DesA1* and *DesA1*

General remarks

Understanding the host pathogen interactions is fundamental and this work is exploring the transcriptional profile of Mtb within an in vivo host cell which has never been performed before. The methodology is a technical advance and therefore will be of interest to not just Mtb researchers but anyone exploring host/pathogen interactions and also could have utility in single cell sequencing (discussed above). The advancement of knowledge as this paper is currently written is more limited. However I don't think (I could be wrong) anyone has ever transcriptionally profiled Mtb growing in macrophages isolated from murine BAL samples but the authors don't capitalise on this (discussed below) and the data is inadequately mined.

Major points

L112-126: It is very difficult to understand what has actually been done here as it is buried in all the numbers.

(a) How many biological replicates of the data are there for the in vivo work? Is it 30 pooled mice three times ie a total of 90 mice? The figure S1A is also unclear (?33 mice? ?X 3=99 (9 mice missing?). Also the data in Table S1 only lists 2 replicates. What happened to the third replicate? These two replicates seem enormously variable (just by eyeballing and performing a quick t-test). What was the variability in the biological replicates (if this data is from biological replicates?)

(b) Why did the authors exclude the genes with zero counts? Do we presume that this RNA was not captured or was there no expression? What is the justification for this? Please clarify in text.

(c) 62% of the genome was identified for comparison but actually only 20% of the genome was actually analysed. What were the results from the analysis with the 62% or is this just to validate the methodology? Please discuss these results.

(d) All the data should be presented in the supplementary datasets as currently if a gene was absent the reader is unsure whether it wasn't differentially expressed, wasn't enriched or was removed because the value was zero. This is a missing part of the story.

(e) I couldn't find anywhere how the Mtb for the in vitro controls were grown. This is significant as this magical media grew Mtb with a similar transcriptional profile to ex vivo grown Mtb. Was Mtb grown in RPMI media for the controls? Is the signature we see as a result of slow growth? (a comparison with other published data is required here).

Generally the data isn't mined sufficiently in the manuscript. The authors need to look at a gene ontology to see if any functional groups are up/down regulated. They immediately look at the most upregulated genes (can often be just very sensitive to change I always think of HSPX) which are two genes that the researchers have already published on. This is important but what does this other exciting data tell us about Mtb growing in the host?

What is the utility of the ex vivo data? Is this more method validation as it is relatively easy to

transcriptional profile Mtb growing in macrophages by RNA-seq. In fact the data presented in this manuscript is highly similar to previously published work. Should this be moved to the method validation? What is new here? Again the authors discuss DesA1 and DesA2 as being highly upregulated. Has this been reported previously? More comparison are required with other published data here.

LL182: The authors identify 59 genes as differentially regulated ex vivo and in vivo but we are not told what these genes are. They should be listed as they could be very interesting!

The systems level analysis of the TF involved is really interesting but the results are slightly strange in that so few networks were identified in the in vivo model. Can the authors speculate why? Is the in vivo environment so neutral that nothing much is happening? They do discuss the recent David Russell paper briefly but this was Mtb after a more established two week infection in mice.

The addition of data about the expression of the deses in a hypoxic module suddenly appearing in the conclusion was generally upsetting and a little bit random. This data should be moved into the main part of the manuscript (see also see comments below about DesA1 and DesA2).

Generally the work on DesA1 and DesA2 is really interesting but feels slightly shoe horned into the this work. A further justification is needed here as to why these two genes are specifically focused on. Indeed DesA1 and DesA2 were not identified in the regulons and therefore the authors had to use another method to identify the regulator. Perhaps this data which is both novel and very interesting should actually be submitted separately in another publication (to a different journal)?

Figure 4 which presents regulation of DesA1 and DesA2 by Rv04272c however many of these genes were not regulated in the in vivo data which needs commenting on in the manuscript

Minor points

The paper is generally very well written and presented but minor points are listed below

L89: To validate "that" is missing

Supplementary tables need titles and legends sufficient for the reader to understand them. What is Rvns? What are these at the beginning MT_H37RV_V3(ng)100? These should also include all data not just the significant genes.

Why is Figure S7 data on Mtb in the supplementary data?

Time scale for major revisions: 3 months to allow additional data mining using gene ontology and other published datasets and a remodelled manuscript

Reviewer #2:

This is an interesting paper that combines the use of novel gene expression analytical methods with systems analysis of the resulting data and confirmatory experimental approaches. Even though the methodologies developed by the authors still do not appear to be robust enough to fully investigate mycobacterial gene expression in the context of the infection of a living animal, the path-seq technique reported here allows a more in-depth interrogation of the bacterial transcriptome in macrophages than previous techniques. Thus, it constitutes an important contribution to tuberculosis research and is in principle applicable to other intracellular pathogens. Combining the computational analysis with the experimental data strengthens both approaches and delivers convincing new biology on two fatty acid desaturases and the TF that regulates their expression.

Minor weaknesses:

1. the writing is very dense, it seems written for a computational audience, and may not be as attractive for experimental biologists.
2. it would be appropriate to cite all the references utilizing similar, TF-based approaches to the analysis of the Mtb transcriptome, including Balazsi et al (PMID: 18985025), which is published in the same journal.

Reviewer #3:

In this work, the authors introduce a method (Path-seq) for transcriptome sequencing and use it to analyze the gene expression of the bacterial pathogen *M. tuberculosis* (Mtb), both during in vivo mouse and in vitro macrophage infections. Path-seq relies on biotinylated oligonucleotide baits to enrich the fraction of bacterial RNA that can be recovered from infected macrophages.

The authors first assess the ability of Path-seq to reliably enrich for bacterial transcripts. They then

analyze the bacterial transcriptional changes associated with in vivo mouse and in vitro macrophage infections. The authors focus a large portion of the analysis on the genes *desA1* and *desA2*, which they find to be upregulated at 24 hrs. after in vivo infection and at 2 hrs. after in vitro infection. They then use computational modeling techniques to characterize the transcription network components associated with infection.

From one such analysis (based on NetSurgeon algorithm), they obtain a set of transcription factors exhibiting significantly high (or low) activity at each time point in the in vivo and in vitro infection data. In a second computational analysis (based on the EGRIN model), they focus on identifying a transcriptional regulator(s) of *desA1* and *desA2*, which they find to be Rv0472c. The authors finally experimentally test the effect of over-expressing the predicted regulator Rv0472c on 1) expression of *desA1* and *desA2*, 2) viability, and 3) mycolic acid biosynthesis in the related bacteria *M. smegmatis*, BCG, and in *Mtb*.

Major comments are below:

The paper revolves around the genes *desA1* and *desA2* and their regulation, but the argument/story that the authors build to validate this focus is at times specious. The main claim is that "Path-seq has led to discovery that MTB transcriptionally regulate (sic) mycolic acids during infection of host cells, influencing virulence and persistence of the pathogen." The logical steps built to support this claim are that:

1.) *desA1* and *desA2* are among the most upregulated genes at 24 hrs. after in vivo infection. The SI data shows that *desA2* is the 7th most up-regulated; *desA1* is the 27th most upregulated gene;

2.) Previous work by the authors and others has shown that the two genes are essential in *Mtb* as demonstrated by in vitro Tn-Seq screens and that depletion (of *desA1*) resulted in reduced viability and mycolic acid biosynthesis in *M. smegmatis*;

3.) *desA1* and *desA2* are transiently upregulated at 2hrs. after in vitro macrophage infection. In fact *desA2* is the 405th most upregulated gene at 2hrs.; and *desA1* is below the 2-fold differential expression cut-off in this data;

4.) The authors claim the NetSurgeon algorithm is a powerful way to reveal which transcription factors' activities are responsible for the observed gene expression changes during both in vitro and in vivo infection. However, none of the identified regulons that come out of the NetSurgeon analysis include *desA1* and *desA2*, and the authors claim that this is probably due to thresholds that were used to avoid false positives.

5) Therefore a different computational approach, based on the EGRIN algorithm, is required to reveal the underlying TF regulation for these two genes. This approach reveals that *desA1* and *desA2* are part of a module that is predicted to be regulated by Rv0472c. Note that the EGRIN model approach does not use the Path-seq data as input, but rather it's "built from a compendium of 2,325 transcriptoms". A more intuitive and simpler explanation of this offered by the authors is that when (in previous work) Rv0472c "was overexpressed in MTB, the TF led to significant repression of 15 genes, but only *desA1* and *desA2* had significant binding of Rv0472c in their promoter region from ChIP-seq analysis".

6) The authors follow up on this last finding by showing that A) in *M. smegmatis*, overexpression of the homologue TF leads to binding to the promoter regions of *desA1* and *desA2* and to their significant repression, B) reduction in viability as measured by CFU counts in *M. smegmatis*, BCG, and *Mtb* (for some reason, the authors decide to put only the *Mtb* results as a supplementary figure); C) in *M. smegmatis*, overexpression of the TF leads to significant changes in metabolic markers of mycolic acid biosynthesis.

Therefore, in my opinion, it's not accurate to say that the Path-seq data has led to the discovery of this mycolic acid regulatory module (i.e. why focus on these two genes and not any in the set of common significantly differentially expressed genes between the two infection models?). Rather the authors interest in *desA1* and *desA2* has led to them running several different analyses around their

expression and regulation. It's perfectly valid to write a paper around *desA1* and *desA2*; but it seems misleading to present it them as the genes and TF that the Path-seq data naturally lead to.

More minor but significant comments are as follows:

The amount of methodological detail that is provided for the transcription network analyses, and the supplementary data that is made available (in the form of supplementary spreadsheets) is incomplete. For instance, since the adaptation of the NetSurgeon algorithm (which was originally designed for transcriptomic/metabolic engineering applications) to this problem is creative and potentially powerful, more details on the implementation should be provided in the Methods section (or in a Supplementary text): what is the basic idea behind NetSurgeon and how is it adapted here (i.e. what are the "initial" and the "goal" transcriptional state?) How are the q-values from the differential gene expression data used to weigh the different genes? More details on the "empirical analysis in *E. coli*" used to set a threshold in the algorithm (which by the way, it's not clear if this threshold refers to the 15 top TFs or the maximum of five targets considered)? These details might be obvious to the authors who are familiar with NetSurgeon, but a brief explanation will strengthen the manuscript.

The manuscript does not place Path-seq in the context of previous closely related works. A literature search revealed that previous publications have used biotinylated oligonucleotides to enrich for pathogenic organisms' transcripts (also isolated from infected hosts) in RNA-seq applications. Relevant references include the following: PMID: 26834721, PMID: 26396240. In addition, there are several papers that do not do RNA-seq, but still perform the biotinylation enrichment trick to study intracellular pathogen transcriptomics: PMID: 10500215, PMID: 10496884. It would be fair to properly cite these and similar works.

The TF network that is used to implement NetSurgeon should be made available as a supplementary data file. I understand that it is based on the network in [PMID: 25380655] (a paper which has a very complete set of SI Tables). But the authors claim they expanded and modified this TF network using a set of operon annotations and by filtering "interactions that did not change at least 20%". Without the updated TF network, replicability of the results is compromised. In general, the standards set by [PMID: 25380655] in terms of supplementary data are, in my opinion, significantly better. Ideally, code used for the EGRIN and NetSurgeon approaches would also be made available.

In the main text, Fig. 1 is referenced in the context of the "mock infection" experiments used to validate Path-seq. However, Fig. 1a is a schematic diagram of a real infection experiment, while Fig. 1b and Fig. 1c are data from the mock infection. Thus the figure is confusing (the reader think that the data in Fig. 1B and 1C comes from pathogen infected host cells, not host RNA spiked with bacterial RNA).

Fig. 2A shows data for the relative expression levels of *desA1*, *desA2*, and *Rv0472c* (*madR*). However, *madR* is introduced much later in the main text than when Fig. 2 is referenced (it is introduced in line 265, when Fig. 4 is first referenced). This is confusing: the reader who just looks at Fig 2 (or who even reads the main text related to Fig 2 and then looks at Fig 2) doesn't know why the *Rv0472c* data is there.

Other minor comments:

The term tSNE is never introduced. Many readers may not be familiar with tSNE and a brief description and/or reference would help.

Are the authors aware that there is a bioinformatics tool out there also called PathSeq ("for the identification and analysis of microbial sequences in high-throughput human sequencing data")? Was the same name (except for the hyphen) chosen on purpose or by chance?

I could be wrong but, in line 110, when the authors say that they obtain a 1000-fold enrichment (in reads that align to MTB), shouldn't this be a 10,000-fold enrichment? A spike of 0.005% MTB RNA that leads to ~30-40% MTB reads is closer to 10,000-fold than to 1,000-fold.

Summary

We are grateful to the referees for their constructive comments and were overall pleased with the favorable reviews. The major criticism was related to the presentation of the data. We have extensively rewritten the manuscript to better substantiate the characterization of DesA1/2, from the *in vivo* Path-seq data. There were numerous lines of reasoning for investigating the regulatory control of *desA1/2* (detailed in responses below) and believe highlighting these points has greatly strengthened the manuscript. Furthermore, the revised manuscript more convincingly demonstrates the powerful combination of the Path-seq data with regulatory network analyses. Together, they were able to deliver convincing new biology and discovery of mycolic acid desaturase regulator (MadR) that likely plays a role in remodeling the cell wall during infection.

Reviewer #1**Major comments:**

1. ***L112-126: It is very difficult to understand what has actually been done here as it is buried in all the numbers.***

We agree with the reviewer - that this section had a lot of detail and numbers. We removed some information that was not pertinent to the story and have clarified how the *in vivo* infection was performed and evaluated. Main text L112-124 now reads:

RNA extracted from AMs in BAL of 10 mice yielded ~100 µg of total RNA. Therefore, we evaluated the Path-seq method using 0.3 µg of BMDM RNA spiked with 0.005% MTB RNA, as an estimate of an *in vivo* infection. We performed Path-seq with two replicates and alignment analysis revealed the percentages of reads that aligned to MTB were 38% and 27%, an approximate 1,000-fold enrichment.

After evaluating the Path-seq methods feasibility for *in vivo* MTB transcriptome analysis, we used flow cytometry to isolate AMs (average of 4.3% of all cells and 83.1% of live, CD45+ cells) in BAL of 30 mice, 24 h after infection with wild-type MTB (**Appendix Figure S1**). Infection, FACS-sorting, and RNA extraction was repeated with three independent mouse infections (three biological replicates), yielding an average ~300 µg total RNA per replicate. The Path-seq enrichment was performed and resulted in 17%, 8% and 5% of the entire reads aligning to MTB for each of the biological replicates.

2. ***How many biological replicates of the data are there for the in vivo work? Is it 30 pooled mice three times ie a total of 90 mice? The figure S1A is also unclear (?33 mice? ?X 3=99 (9 mice missing?).***

We have clarified the number of biological replicates for the *in vivo* work (three biological replicates) in the main text. Additionally, we have clarified in Appendix Figure S1A that three mice from each biological replicate (9 mice total) were used to evaluate the deposition during each infection.

Main text L120-124 now reads:

Infection, FACS-sorting, and RNA extraction was repeated with three independent mouse infections (three biological replicates), yielding an average ~300 µg total RNA per replicate. The Path-seq enrichment was performed and resulted in 17%, 8% and 5% of the entire reads aligning to MTB for each of the biological replicates.

Additionally, the figure legend of Appendix Figure S1A now reads:

Schematic of *in vivo* infection. Infection, FACS-sorting and RNA extraction was repeated with three independent mouse infections. At 24 h post infection, the lungs of three mice were collected and plated for colony forming units (CFUs) to evaluate deposition of each infection (deposition reported in **Appendix Table S1**). Bronchoalveolar lavage (BAL) was performed on the remaining 30 mice, followed by FACS isolation of alveolar macrophages from BAL and RNA extraction. Path-seq was performed on all three replicates.

3. ***Also the data in Table S1 only lists 2 replicates. What happened to the third replicate? These two replicates seem enormously variable (just by eyeballing and performing a quick t-test). What was the variability in the biological replicates (if this data is from biological replicates?)***

Dataset EV1 gives the normalized read counts for all significantly differentially expressed genes between *in vivo* (columns AM_rep1 and AM_rep2) and extracellular samples (columns control_rep1, control_rep1, and control_rep3). For this analysis, we excluded biological replicate three as only ~14% of genes had non-zero values and the total read count was significantly less than the other replicates (summarized in **Appendix Table S1**). The deposition in replicate 3 was also lower, suggesting that there was lower amount of starting MTB and presumably not captured as efficiently in biological replicate 3. We believe that variability between biological replicates is not an issue for the differential expression analysis, given we restricted to transcripts with non-zero read counts in both replicates. In fact, despite the variability between mice replicates, the majority of genes classified as differentially expressed in the *in vivo* model show distinct expression patterns between control and mice samples. The variability between the *in vivo* samples (columns AM_rep1 and AM_rep2) is taken into account by DESeq2 when calculating the statistical significance (q-values presented in **Dataset EV1**) of changes in expression between *in vivo* and extracellular samples. To do so, DESeq computes and incorporates gene-wise dispersion estimates and shrinkage of fold changes in the pipeline. That is one of the reasons why DESeq2 has been widely adopted for analyses with limited number of replicates. We clarified the exclusion of the third replicate in the main text L135-143:

To calculate differentially expressed genes between the *in vivo* and extracellular samples, we excluded *in vivo* biological replicate 3, which had significantly lower total read counts compared to all other samples (summarized in **Appendix Table S1**). The lower deposition of replicate 3 suggests a lower amount of starting MTB could explain the low read count, as opposed to a difference in gene expression. Differential expression analysis between *in vivo* intracellular MTB and extracellular MTB, identified 431 significantly differentially expressed genes (\log_2 fold change < -1.0 or > 1.0 and multiple hypothesis corrected P -value < 0.05 , **Dataset EV1**).

4. ***Why did the authors exclude the genes with zero counts? Do we presume that this RNA was not captured or was there no expression? What is the justification for this? Please clarify in text.***

The reviewer brings up a valid point that was not discussed in the text. We suspect the lower starting amount of MTB RNA (compared to host RNA) affects the capture in the *in vivo* samples. This results in a larger number of genes with zero counts, and is not a reflection of reduced gene expression. As such, we have erred on the side of caution in evaluating the comprehensiveness of Path-seq used *in vivo* and excluded genes with zero counts across all *in vivo* replicates. It is worth noting that only ~2% of genes have zero values from the extracellular samples (summarized in **Appendix Table S1**), also processed with Path-seq, and therefore there is no reason to believe it is an artifact of the probe library. Furthermore, we have made available the raw data and normalized reads for ALL genes (with zero reads) in GEO dataset GSE116394. We have clarified these points in the main text L129-136:

We suspect genes with non-detectable reads are a result of the miniscule amount of MTB RNA compared to host RNA in the *in vivo* samples, and not a reflection of real gene expression changes. Therefore, excluding genes with zero counts in all *in vivo* replicates resulted in 3,505 MTB genes (62% of genome) with sequenced expression measurements from *in vivo* infection using Path-seq. These results (raw data and normalized read counts for ALL genes are available in GEO: GSE116394) present the most comprehensive transcriptome profiling of MTB from *in vivo* infection and a major technical advancement for researchers studying *in vivo* host-pathogen interactions.

5. ***62% of the genome was identified for comparison but actually only 20% of the genome was actually analysed. What was the results from the analysis with the 62% or is this just to validate the methodology? Please discuss these results.***

The 62% of the MTB genome represents the comprehensiveness of Path-seq method in the *in vivo* conditions used in this study. We have since performed Path-seq on AMs isolated 20 days after infection and have genes with non-zero values comparable to the extracellular samples (~98% of the genome). While we do not include these results in this manuscript, we have clarified the significance of the 62% of the genome in main text L129-136:

We suspect genes with non-detectable reads are a result of the miniscule amount of MTB RNA compared to host RNA in the *in vivo* samples, and not a reflection of real gene

expression changes. Therefore, excluding genes with zero counts in all *in vivo* replicates resulted in 3,505 MTB genes (62% of genome) with sequenced expression measurements from *in vivo* infection using Path-seq. These results (raw data and normalized read counts for ALL genes are available in GEO: GSE116394) present the most comprehensive transcriptome profiling of MTB from *in vivo* infection and a major technical advancement for researchers studying *in vivo* host-pathogen interactions.

6. ***All the data should be presented in the supplementary datasets as currently if a gene was absent the reader is unsure whether it wasn't differentially expressed, wasn't enriched or was removed because the value was zero. This is a missing part of the story.***

We completely agree with the reviewer and have made available the raw data and normalized reads for ALL genes (including those with zero reads) in GEO dataset GSE116394. The supplementary datasets contain the significantly differentially expressed genes, as calculated with our methods and significance thresholds (described in materials and methods). The normalized reads available in GEO allow researchers to make their own interpretation of differentially expressed genes. We have clarified this in the main text text L133-136:

These results (raw data and normalized read counts for ALL genes are available in GEO: GSE116394) present the most comprehensive transcriptome profiling of MTB from *in vivo* infection and a major technical advancement for researchers studying *in vivo* host-pathogen interactions.

7. ***I couldn't find anywhere how the Mtb for the in vitro controls were grown. This is significant as this magical media grew Mtb with a similar transcriptional profile to ex vivo grown Mtb. Was Mtb grown in RPMI media for the controls? Is the signature we see as a result of slow growth? (a comparison with other published data is required here).***

We thank the reviewer for pointing out this oversight and have clarified the growth conditions for the extracellular controls in both the main text and materials and methods.

Main text L124-126:

We compared the MTB read counts between the *in vivo* samples with extracellular samples, obtained from MTB grown in 7H9 media for 24 h (starting OD₆₀₀ = 0.1). Both the *in vivo* and extracellular samples were processed by Path-seq.

Materials and methods 794-795:

The same MTB H37Rv cultures used for infection were also diluted to starting OD₆₀₀ = 0.1 and grown in 7H9 media.

8. ***Generally the data isn't mined sufficiently in the manuscript. The authors need to look at a gene ontology to see if any functional groups are up/down regulated. They immediately look at the most upregulated genes (can often be just very sensitive to change I always think of HSPX) which are two genes that the researchers have already published on. This is important but what does this other exciting data tell us about Mtb growing in the host?***

We agree with the reviewer that we neglected to fully mine the *in vivo* Path-seq data. We thank the reviewer for this comment and believe the additional analyses and clarification (throughout the main text) of why DesA1/2 were selected for further characterization have really strengthened the manuscript. Please see response to **comment 13** for further detail on the points we have clarified throughout the manuscript. Furthermore, as the reviewer suggested, we performed functional enrichment analysis on the *in vivo* significantly enriched genes. We also further describe why the up-regulation of the specific mycolic acid biosynthesis genes with AMs is particularly interesting. We more comprehensively discuss the *in vivo* data in the main text L144-168 as following:

Among the differentially expressed transcripts that code for annotated proteins (376 genes), 121 were down-regulated and significantly enriched (multiple hypothesis corrected *P*-value = 0.005, in the proline-glutamic acid (PE)/proline-proline-glutamic acid (PPE) family of proteins. The exact physiological role of the PE and PPE proteins in MTB is yet to be fully understood, but they are thought to play important roles in immune evasion (Tiwari et al, 2012). It is interesting that PE and PPE genes were down-regulated in AMs and might indicate state they are unnecessary within these host cells. In addition, 255 genes were up-regulated in AMs at 24 h post infection and were significantly enriched (*P*-value < 0.05) in the functional categories: 'insertion sequences and phages', 'information pathways', and 'lipid metabolism' (*P*-value = 0.002). Most interesting, many of the genes whose protein

products are associated with ‘lipid metabolism’ are involved in the biosynthesis of mycolic acid. These up-regulated mycolic acid biosynthesis genes include *umaA*, *pcaA*, *desA1*, *desA2*, *fadD32* and *fabD*. In addition, are genes of the operon involved in phthiocerol dimycocerosate (PDIM) biosynthesis. The biosynthesis of mycolates is an energetically expensive process and found to be repressed in MTB upon entry into dormancy (Galagan et al, 2013, Jamet et al, 2015). This suggests that MTB in AMs 24 h post infection are not in a dormant state. Instead, the transcriptional response indicates that MTB is actively remodeling the cell wall, perhaps with modifications that specifically contribute to survival within AMs. Interestingly, the up-regulated genes, *umaA* and *pcaA*, are required for synthesis of cyclopropane ring formation in mycolic acids of MTB. Furthermore, *desA1* and *desA2* (with log₂ fold change of 4.0 and 4.7, respectively within AMs) encode fatty acid desaturases that introduce double bonds into fatty acids (Singh et al, 2016). Desaturation is a necessary step prior to cyclopropanation and other mycolic acid modifications. It is interesting to speculate that conditions within AMs induce desaturation events, enabling MTB to fine-tune subsequent cyclopropanation and other modifications of mycolic acids that contribute to cell wall permeability and adaptation within these host cells. The significant up-regulation of mycolic acid remodeling genes following *in vivo* infection was interesting and deserved further investigation of their transcriptional control.

9. ***What is the utility of the ex vivo data? Is this more method validation as it is relatively easy to transcriptional profile Mtb growing in macrophages by RNA-seq. In fact the data presented in this manuscript is highly similar to previously published work. Should this be moved to the method validation? What is new here? Again the authors discuss DesA1 and DesA2 as being highly upregulated. Has this been reported previously? More comparison are required with other published data here.***

The *ex vivo* data (termed ‘*in vitro*’ in the manuscript and henceforth in response) was used to profile the differentially expressed *in vivo* genes, specifically those involved in mycolic acid biosynthesis, across a time course of infection. The large number of mice prohibited *in vivo* infection time course. While the reviewer states it is relatively easy to transcriptionally profile MTB from *in vitro* macrophages, this has only been done with microarrays and in a limited number of studies, to our knowledge (Schnappinger et al, 2003, Rohde et al, 2007, Rohde et al, 2012). Here, we used sequence-based transcriptome profiling which adds to the transcript features that can be measured and we simultaneously capture gene expression from both host and pathogen. We agree with the reviewer that the reasoning for the *in vitro* infection data was poorly explained and have extensively revised the main text. We also directly compared the expression of *desA1/2* and other mycolic acid biosynthesis genes to previously published work (only *umaA*, *desA1* and *desA2* were found to be up-regulated in the *in vitro* infection microarray analyses). Moreover, we added **Appendix Table 2**, which details the overlap of all differentially expressed genes between our *in vitro* Path-seq dataset and previously published microarray datasets.

We further describe the utility of the *in vitro* infection data and distinguish our work from previously published work in the main text L171-191 as following:

Several genome-wide expression studies of MTB challenged with dormancy-inducing stresses, such as nutrient starvation (Jamet et al, 2015) and hypoxia (Galagan et al, 2013, McGillivray et al, 2015), have shown that genes involved in mycolic acid biosynthesis are generally down-regulated. In contrast, we observed up-regulation of mycolate biosynthesis genes in MTB from AMs of infected mice at 24 h. Therefore, we sought to study the expression of these mycolic acid modification genes at multiple time points during infection using the Path-seq method and MTB infected bone marrow derived macrophages (BMDMs). An *in vitro* infection system was used due to the large number of mice required for additional time points during *in vivo* infection. We isolated murine BMDMs and infected them with MTB at a MOI of 10. Infected cells were collected at 2, 8 and 24 h after infection along with extracellular MTB grown in 7H9 media as control. Total RNA was extracted, depleted of rRNA and handled as described above (**Fig. 1A**). All extracellular MTB samples were processed by Path-seq as well. For the *in vitro* infection samples (**Appendix Figure S2**), we split each sample into RNA-seq and Path-seq fractions to evaluate the enrichment efficiency and to simultaneously obtain both host and pathogen transcriptomes from the same infection sample. While we did not perform transcriptome

analysis of the host cells in this study, the raw data is available (GSE116357) along with uninfected BMDM controls, and is the first dual monitoring of both MTB and host transcriptomes from the same infection samples. The percentage of reads that aligned to MTB was consistent at a 100-fold increase in the enriched vs nonenriched samples across replicates and time points (Table 1). With an average 11 million (M) mapped reads for both intracellular (average 13.4 M) and extracellular (average 8.8 M) MTB, we obtained >100x coverage and 5,622 unique features (including ncRNA, UTRs, etc.). This is further validation of the Path-seq method to comprehensively study the authentic intracellular state of a pathogen.

10. LL182: The authors identify 59 genes as differentially regulated ex vivo and in vivo but we are not told what these genes are. They should be listed as they could be very interesting!

We completely agree with the reviewer and have added **Extended View Dataset 3** that includes genes that were significantly differentially expressed in both models of infection. The dataset distinguishes genes that were down- or up-regulated in both models or were incongruously expressed between the models. This dataset also includes functional enrichment of the commonly up-regulated genes. We have also revised **Appendix Figure S4** to include common differentially expressed genes across the models from any time point, not just 24 h. We have expanded on the comparison between the two models in the main text L233-240 as following:

In addition to the mycolic acid biosynthesis gene, we further compared all significantly differentially expressed genes between the two infection models and found only a small but significant (P -value < 0.01) subset of common genes (59 genes at 24 h and 137 genes from any *in vitro* time point, **Dataset EV3** and **Appendix Figure S4**). Most of the common genes were up-regulated in both models and included genes significantly enriched in categories related to the ribosome and response to hypoxia according to MTB annotation in DAVID (Huang da et al, 2009, Huang da et al, 2009). Interestingly both AAA+ ATPases, *clpX* (Rv2457c) and *clpCI* (Rv3596c), that interact with the ClpP proteolytic core (Neuwald et al, 1999, Raju et al, 2014), were also significantly up-regulated in both models.

11. The systems level analysis of the TF involved is really interesting but the results are slightly strange in that so few networks were identified in the in vivo model. Can the authors speculate why? Is the in vivo environment so neutral that nothing much is happening? They do discuss the recent David Russell paper briefly but this was Mtb after a more established two week infection in mice.

This is an interesting observation from the reviewer and we agree that it deserves an explanation. We note that the NetSurgeon method can be impacted by the type of differentially expressed genes. For example, if the *in vivo* differentially expressed genes were enriched in genes that do not belong to regulons. However, we don't find that to be the

case. Based on the Path-seq data, we believe the environments within these different host cells present different cues to MTB, thus impacting gene expression. The systems-level analysis confirms different active regulatory networks and that MTB within AMs were lacking the stress networks that

were active within BMDMs. **Response Figure 1** demonstrates a regulon (controlled by KstR) that is present in both datasets (*in vitro* and *in vivo*), but KstR only has decreased activity in BMDMs. Another possible explanation could be that heterogeneity of infection (timing, host-pathogen interaction, etc.) could be greater *in vivo* and thus impact the number of active regulatory network identified by NetSurgeon. We have included our speculative explanation of these differences in the main text L306-310 as following:

Overall, there were far fewer active networks identified *in vivo*, compared to the *in vitro* infection. While the type of differentially expressed genes (i.e. genes not belonging to regulons) could contribute to such differences, we do not see that being the case. Therefore, it is appealing to speculate that the more permissive environment within AMs and a greater heterogeneity of infection from the *in vivo* model could contribute to the differences in the number of active regulatory networks identified by NetSurgeon.

12. The addition of data about the expression of the deses in a hypoxic module suddenly appearing in the conclusion was generally upsetting and a little bit random. This data should be moved into the main part of the manuscript (see also see comments below about DesA1 and DesA2).

We see the reviewers point and have moved this data and related Figure to the main text, in the revised section L214-230, with section title '***desA1 and desA2 are induced early during in vitro macrophage infection and hypoxia time***'.

13. Generally the work on DesA1 and DesA2 is really interesting but feels slightly shoe horned into the this work. A further justification is needed here as to why these two genes are specifically focused on. Indeed DesA1 and DesA2 were not identified in the regulons and therefore the authors had to use another method to identify the regulator. Perhaps this data which is both novel and very interesting should actually be submitted separately in another publication (to a different journal)?

The reviewer appropriately critiques an unexplained focus on DesA1/2 from the *in vivo* Path-seq data. We have done a major rewrite of the manuscript that better justifies the detailed characterization of the regulatory control of *desA1/2*. The major points that support this are:

- a) We performed functional enrichment of the differentially expressed genes and found significant enrichment of 'lipid metabolism' associated genes from the *in vivo* Path-seq data. Among the genes in this category, included a large number of genes involved in mycolic acid biosynthesis and specifically modifications to mycolates.
- b) The up-regulation of mycolic acid biosynthesis genes in MTB from AMs was particularly interesting given that the synthesis of mycolates is an energetically expensive process and generally down-regulated in stressful environments.
- c) We further explored the expression of these mycolic acid biosynthesis genes during *in vitro* infection and found only *umaA*, *desA1/2* were significantly differentially expressed at any time point. The desaturases were transiently up-regulated at 2 h, then returned to normal levels, while *umaA* was up-regulated across all time points.
- d) Furthermore, *desA1/2* were initially up-regulated and then subsequently down-regulated over a hypoxia time course, suggesting their tight regulatory control. *UmaA* was not differentially expressed at any oxygen concentration across the time course.
- e) Given the dynamic expression of *desA1/2* during infection and low oxygen stress, we used systems-level approaches to identify their regulator(s). The EGRIN model

predicted co-regulation of the desaturases by Rv0472c, which was supported by data from the Rv0427c over-expression strain.

The above points led to the validation and characterization experiments that are described in the manuscript. We have done extensive editing of the manuscript to better describe these points above. We believe that the inclusive manuscript demonstrates how combining the Path-seq experimental method with network analyses strengthens both approaches. Together, they were able to deliver convincing new biology on the regulatory control of fatty acid desaturases that likely play a role in remodeling the cell wall during infection.

14. Figure 4 which presents regulation of DesA1 and DesA2 by Rv04272c however many of these genes were not regulated in the in vivo data which needs commenting on in the manuscript

The reviewer brings up a valid point that was not discussed in the text. We believe the reviewer is commenting on the other genes (besides *desA1* and *desA2*) that were predicted to be co-regulated in bicluster 276 (shown in Figure 4). In fact, a significant number of the genes in bicluster 276 were up-regulated in the *in vivo* Path-seq data (now described in the text and shown in Figure 4). However, what is unclear is that Rv0472c only regulates *desA1* and *desA2* within bicluster 276. Interestingly, the other bicluster genes are involved in the biosynthesis/transport of PDIM (component of mycobacteria cell wall). We believe this demonstrates a unique feature of EGRIN biclusters, which are capable of bringing together functionally related genes from different regulons. We speculate that bicluster 276 presents a network of genes that collectively alter cell wall composition (permeability, rigidity, etc.) during infection. We have included these points in the main text L363-373 as following:

This analysis demonstrates the utility of EGRIN to identify the regulator of genes relevant to infection. Yet, the Rv0472c and MSMEG_0916 overexpression data supports the direct regulation of only *desA1/2*, among bicluster 276 genes. It is worth noting that EGRIN biclusters can be overlapping sets of co-regulated genes that, in some cases, group together genes from different regulons, and in other cases, subdivide genes of the same regulon, or even the same operon. While a significant number of bicluster 276 genes are up-regulated during *in vivo* infection, not all of the genes are necessarily regulated by the same TF. As such, bicluster 276 represents a coordination of regulatory mechanisms that bring together functionally related genes. These genes, involved in biosynthesis/transport of PDIM and desaturation of mycolic acids, act together to alter cell wall composition, thereby affecting cell wall permeability and host responses during *in vivo* infection.

Minor comments:

The paper is generally very well written and presented but minor points are listed below

1. L89: To validate "that" is missing

We thank the reviewer for this attention to detail. We have fixed the main text as following:

Accounting for BMDMs **that** might not be infected and based on intracellular sequencing studies from literature

2. Supplementary tables need titles and legends sufficient for the reader to understand them. What is Rvns? What are these at the beginning MT_H37RV_V3(ng)100? These should also include all data not just the significant genes.

We completely agree with the reviewer and have edited the titles and legends in the Appendix for better understanding. We believe the reviewer is referring to the non-coding transcripts that are included in the significantly differentially expressed genes found in the Extended View Datasets. We have included a column for protein product in these datasets, which includes a description of these transcripts (and all transcripts). As mentioned in comment 6, we have made available the raw data and normalized read counts for ALL transcripts in GEO dataset GSE116394. The supplementary datasets contain the significantly differentially expressed genes, as calculated with our methods and significance thresholds (described in materials and methods). The normalized read counts available in GEO allow researchers to easily make their own interpretation of differentially expressed genes, using their preferred method.

3. Why is Figure S7 data on Mtb in the supplementary data?

We see the reviewers point and the significance of the MTB viability data. Thus, we have moved description of the MTB data and related Figure to the main text, L381-388 and Figure 5, and MSM data to the Appendix.

Plates with MTB and *Rv0472c* overexpression strain were incubated for 3 weeks and growth patterns indicated that the presence of ATc resulted in a 4-log fold reduction in CFU counts (**Fig. 5A**). In comparison, plates containing the parental MTB strain showed no change in CFUs with the presence or absence of ATc (**Fig. 5A**). Similar experiments were done in *Mycobacterium bovis* BCG (BCG) and MSM with *Rv0472c* and *MSMEG_0916* overexpression, respectively. Overexpression resulted in 3-log viability reduction in BCG and (**Appendix Figure S6A**) and 2-log viability reduction in MSM (**Appendix Figure S6B**). We also observed very limited growth in broth culture when *MSMEG_0916* was induced with ATc (**Appendix Figure S7**).

Reviewer #2

Major comments:

This is an interesting paper that combines the use of novel gene expression analytical methods with systems analysis of the resulting data and confirmatory experimental approaches. Even though the methodologies developed by the authors still do not appear to be robust enough to fully investigate mycobacterial gene expression in the context of the infection of a living animal, the path-seq technique reported here allows a more in-depth interrogation of the bacterial transcriptome in macrophages than previous techniques. Thus, it constitutes an important contribution to tuberculosis research and is in principle applicable to other intracellular pathogens. Combining the computational analysis with the experimental data strengthens both approaches and delivers convincing new biology on two fatty acid desaturases and the TF that regulates their expression.

We appreciate the reviewer's kind support of our work.

Minor comments:

1. the writing is very dense, it seems written for a computational audience, and may not be as attractive for experimental biologists.

We have extensively edited the manuscript and hope the reviewer finds the writing less dense and understandable to a wide range of scientists.

2. it would be appropriate to cite all the references utilizing similar, TF-based approaches to the analysis of the Mtb transcriptome, including Balazsi et al (PMID: 18985025), which is published in the same journal.

We thank the reviewer for this suggestion. We have included a few references to other systems-level approaches to identify transcriptional networks in the main text L253-255 as following:

There are numerous methods for identifying the key transcriptional networks from different environments (Balazsi et al, 2008, Brynildsen & Liao, 2009, Cahan et al, 2014). Based on one such approach, NetSurgeon (Michael et al, 2016), we evaluated the role of each TF in the observed gene expression changes given a signed transcriptional network.

Reviewer #3**Major comments:**

*In my opinion, it's not accurate to say that the Path-seq data has led to the discovery of this mycolic acid regulatory module (i.e. why focus on these two genes and not any in the set of common significantly differentially expressed genes between the two infection models?). Rather the authors interest in *desA1* and *desA2* has led to them running several different analyses around their expression and regulation. It's perfectly valid to write a paper around *desA1* and *desA2*; but it seems misleading to present it them as the genes and TF that the Path-seq data naturally lead to.*

The reviewer has accurately captured the main points of the manuscript. The reviewer also appropriately critiques an unexplained focus on *DesA1/2* from the *in vivo* Path-seq data. We have done a major rewrite of the manuscript that better justifies the detailed characterization of the regulatory control of *desA1/2*. The major points that support this are:

- a) We performed functional enrichment of the differentially expressed genes and found significant enrichment of 'lipid metabolism' associated genes from the *in vivo* Path-seq data. Among the genes in this category, included a large number of genes involved in mycolic acid biosynthesis and specifically modifications to mycolates.
- b) The up-regulation of mycolic acid biosynthesis genes in intracellular MTB was interesting given that the synthesis of mycolates is an energetically expensive process and generally down-regulated in stressful environments.
- c) We further explored the expression of these mycolic acid biosynthesis genes during *in vitro* infection and found only *umaA*, *desA1/2* were significantly differentially expressed at any time point. The desaturases were transiently up-regulated at 2 h, then returned to normal levels, while *umaA* was up-regulated across all time points.
- d) Furthermore, *desA1/2* were initially up-regulated and then subsequently down-regulated over a hypoxia time course, suggesting tight regulatory control. *UmaA* was not differentially expressed across the oxygen concentrations and time points.
- e) Given the dynamic expression of *desA1/2* during infection and low oxygen stress, we used systems-level approaches to identify their regulator(s). The EGRIN model predicted co-regulation of the desaturases by Rv0472c, which was supported by data from the Rv0427c over-expression strain.

The above points led to the validation and characterization experiments that are described in the manuscript. We have done extensive editing of the manuscript to better describe these points above. We believe that the inclusive manuscript demonstrates how combining the Path-seq experimental method with network analyses strengthens both approaches. Together, they were able to deliver convincing new biology on the regulatory control of fatty acid desaturases that likely play a role in remodeling the cell wall during infection.

Minor comments:

1. *The amount of methodological detail that is provided for the transcription network analyses, and the supplementary data that is made available (in the form of supplementary spreadsheets) is incomplete. For instance, since the adaptation of the NetSurgeon algorithm (which was originally designed for transcriptomic/metabolic engineering applications) to this problem is creative and potentially powerful, more details on the implementation should be provided in the Methods section (or in a Supplementary text): what is the basic idea behind NetSurgeon and how is it adapted here (i.e. what are the "initial" and the "goal" transcriptional state?) How are the q-values from the differential gene expression data used to weigh the different genes? More details on the "empirical analysis in E. coli" used to set a threshold in the algorithm (which by the way, it's not clear if this threshold refers to the 15 top TFs or the maximum of five targets considered)? These details might be obvious to the authors who are familiar with NetSurgeon, but a brief explanation will strengthen the*

manuscript.

We completely agree with the reviewer and have added much more detail to the methods section describing the implementation of NetSurgeon, along with making available the network and R code, for the computational analyses we performed. The revised methods section is pasted below. We have also included gene name and protein product information in the Extended View datasets, which provides a description of the differentially expression transcripts. We have made available the raw data and normalized read counts for ALL transcripts in GEO dataset GSE116394. The Extended View datasets contain the significantly differentially expressed genes, as calculated with our methods and significance thresholds (described in materials and methods). The normalized read counts available in GEO allow researchers to easily make their own interpretation of differentially expressed genes, using their preferred method. The revised methods section L918-947 now reads:

Identification of transcription factors with differential activity in intracellular MTB (using NetSurgeon)

We identified potential TFs with increased or decreased regulatory activity in intracellular MTB (respect to extracellular controls) at each sampled time point using the method recently developed by Michael *et al.* called the NetSurgeon algorithm (Michael et al, 2016). Briefly, NetSurgeon identifies TFs whose differential regulatory activity is likely responsible for the observed transcriptional changes between two states of interest. In our case, we wanted to identify TFs that drive differential expression between intracellular MTB and their controls. Changes in TF activities are estimated based on the expression of their target genes (derived from DESeq2 output). TF regulons are extracted from a signed transcriptional regulatory network specified by the user. The signed MTB transcriptional network model used in this study is described above (and available at <http://networks.systemsbiology.net/mtb>). NetSurgeon's scoring is based on the hypergeometric test distribution (Michael et al, 2016). Three important NetSurgeon's considerations are: i) increase and decrease in TF activity are independently scored; ii) only target genes differentially expressed (according to user's defined *P*-values, *q*-values and fold change cutoffs) in the proper direction impact TF scores. This means that in case of increased activity, only genes significantly down-regulated and up-regulated will contribute to the score of their repressors and activators, respectively; and iii) TF scores are defined not only by the number of target genes that are differentially expressed in the correct direction, but also by their adjusted *P*-values (associated with the differential expression analysis performed with DESeq2). The weight of each gene in the scores of its transcriptional regulators is proportional to the $-\log_2$ of its adjusted *P*-value. Thus, NetSurgeon's score is not homogeneous distributed among TFs' targets but it is affected by the differential expression significance of each gene. To reduce false positives (i.e. misleading presence of TFs with small number of known targets at the top of our TF scores ranking) due to overlap between regulons, only TFs with at least five targets were considered in this analysis. Furthermore, we considered TFs in the top 15 of NetSurgeon's score ranking as the ones with differential activity. This threshold was selected based on the NetSurgeon's results when we compared susceptible and antibiotic resistance *Escherichia coli* transcriptomes. Among the top 15 TFs, we were able to recall TFs known to be involved in multidrug resistance (such as MarA and homolog Rob, Ruiz & Levy, 2010). TFs with less than five targets with differential expression or with contradicting trends (similar number of activated and repressed targets moving in the same direction) were excluded from the NetSurgeon's top 15 ranking. For the *in vivo* and 8 h *in vitro* data, the next two TFs outside the top 15 ranking were included to compensate for multiple TFs that were filtered out.

2. ***The manuscript does not place Path-seq in the context of previous closely related works. A literature search revealed that previous publications have used biotinylated oligonucleotides to enrich for pathogenic organisms' transcripts (also isolated from infected hosts) in RNA-seq applications. Relevant references include the following: PMID: 26834721, PMID: 26396240. In addition, there are several papers that do not do RNA-seq, but still perform the***

biotinylation enrichment trick to study intracellular pathogen transcriptomics: PMID: 10500215, PMID: 10496884. It would be fair to properly cite these and similar works.

The reviewer brings up a good point. We have added these previous studies, and pointed out their differences in the main text L62-71 as following:

To improve the coverage of pathogen transcripts, we made use of biotinylated oligonucleotide baits that are complementary to the pathogen transcriptome. The baits are hybridized to mixed host-pathogen RNA and used to enrich pathogen transcripts for sequencing. **Approaches using biotinylated genome fragments have previously been used to enrich specific genes of intracellular pathogens (Graham & Clark-Curtiss, 1999, Morrow et al, 1999) or perform genome-wide transcriptome profiling of fungal RNA from infected host cells (Amorim-Vaz et al, 2015).** Here, we applied the pathogen-sequencing (Path-seq) method to explore transcriptional changes in MTB **(one-fourth the size of the fungus, *Candida albicans*)** following infection in mice. following infection in mice. Path-seq **data along with network modeling** has led to discovery that MTB transcriptionally regulate mycolic acids during infection of host cells, influencing virulence and persistence of the pathogen.

3. ***The TF network that is used to implement NetSurgeon should be made available as a supplementary data file. I understand that it is based on the network in [PMID: 25380655] (a paper which has a very complete set of SI Tables). But the authors claim they expanded and modified this TF network using a set of operon annotations and by filtering "interactions that did not change at least 20%". Without the updated TF network, replicability of the results is compromised. In general, the standards set by [PMID: 25380655] in terms of supplementary data are, in my opinion, significantly better. Ideally, code used for the EGRIN and NetSurgeon approaches would also be made available.***

We completely agree with the reviewer and have made the network used with NetSurgeon available on the MTB Network Portal <http://networks.systemsbiology.net/mtb/data-center> and the R notebook, with code for input files for NetSurgeon available on the Baliga lab GitHub <https://github.com/baliga-lab> (along with the code for constructing EGRIN models). The MTB EGRIN was previously made easily available for researchers to explore on the MTB Network Portal (Turkarlan et al, 2015), at <http://networks.systemsbiology.net/mtb/>.

4. ***In the main text, Fig. 1 is referenced in the context of the "mock infection" experiments used to validate Path-seq. However, Fig. 1a is a schematic diagram of a real infection experiment, while Fig. 1b and Fig. 1c are data from the mock infection. Thus the figure is confusing (the reader think that the data in Fig. 1B and 1C comes from pathogen infected host cells, not host RNA spiked with bacterial RNA).***

We see the confusion the reviewer brings up. Therefore, we have changed Fig 1a to depict mock infection OR real infection as the starting material in the Path-seq schematic.

5. ***Fig. 2A shows data for the relative expression levels of *desA1*, *desA2*, and *Rv0472c* (*madR*). However, *madR* is introduced much later in the main text than when Fig. 2 is referenced (it is introduced in line 265, when Fig. 4 is first referenced). This is confusing: the reader who just looks at Fig 2 (or who even reads the main text related to Fig 2 and then looks at Fig 2) doesn't know why the *Rv0472c* data is there.***

The reviewer brings up a good point, and an inconsistency in the flow of information in the manuscript. Therefore, we have removed *Rv0472c* (*madR*) from the plots in Fig 2.

Other minor comments:

1. ***The term tSNE is never introduced. Many readers may not be familiar with tSNE and a brief description and/or reference would help.***

The reviewer brings up a good point and oversight. We have introduced the term t-SNE, added reference, and included a brief description in the main text L192-196, which now reads:

Using the normalized read counts from the intracellular and extracellular MTB data, we identified two clusters by implementing the R NbClust function (Charrad et al, 2014) on principal component analysis output, a dimensionality reduction method. The two identified clusters are shown in a two-dimension t-distributed stochastic neighbor embedding plot (t-SNE, van der Maaten & Hinton, 2008) plot.

2. ***Are the authors aware that there is a bioinformatics tool out there also called PathSeq ("for the identification and analysis of microbial sequences in high-throughput human sequencing data")? Was the same name (except for the hyphen) chosen on purpose or by chance?***

We appreciate the reviewer bringing this to our attention and admit the same name was chosen by chance. We previously did an internet search for Path-sequencing and did not find any matches. We have looked into the "PathSeq" mentioned by the reviewer and would like to continue calling our method "Path-seq". Our name and the method it describes are more consistent with other methods generating sequence-based data (i.e. RNA-seq, ChIP-seq, ATAC-seq, etc). In contrast, "PathSeq" is a bioinformatics tool for subtracting human sequences to discover pathogens in potentially infected tissues. Further, we would like to note that in the time that the manuscript has been available on BioRxiv pre-print repository, the manuscript has been very well received and we've already received requests for using Path-seq. We feel the name is already familiar to the research community.

3. ***I could be wrong but, in line 110, when the authors say that they obtain a 1000-fold enrichment (in reads that align to MTB), shouldn't this be a 10,000-fold enrichment? A spike of 0.005% MTB RNA that leads to ~30-40% MTB reads is closer to 10,000-fold than to 1,000-fold.***

We appreciate the reviewer's attention to detail. We were being conservative in stating the enrichment efficiency, as it was not quite 10,000-fold. We agree with the reviewer that it is closer to 10,000 than 1,000-fold enrichment and have therefore edited the main text L112-117 to read the following:

RNA extracted from AMs in BAL of 10 mice yielded ~100 µg of total RNA. Therefore, we first evaluated the Path-seq method using 0.3 µg of BMDM RNA spiked with 0.005% MTB RNA, to simulate mixed host and pathogen RNA composition of a sample from an *in vivo* infection. We performed Path-seq with two replicates and alignment analysis revealed the percentages of reads that aligned to MTB were 38% and 27%, an approximate 10,000-fold enrichment.

2nd Editorial Decision

30th January 2019

Thank you for sending us your revised manuscript. We have now heard back from reviewer #1 who was asked to evaluate your revised study. As you will see below, the reviewer is satisfied with the performed revisions. They list however a few remaining minor issues, which we would ask you to address before we formally accept the study for publication.

REFEREE REPORT

Reviewer #1:

The modified manuscript Path-seq identifies an essential mycolate remodeling program for mycobacterial host adaptation by Paterson et al is a significant improvement on the original submission. The authors have answered all of my comments to my satisfaction. This manuscript represents a significant piece of original work of suitable quality and interest for publication in Molecular Systems Biology subject to the minor corrections listed below.

Minor points

The paper is generally very well written and presented but minor points are listed below –

P4 L156: something missing from this sentence as makes no sense

A1, desA2, fadD32 and fabD In addition, are genes of the operon involved in phthiocerol dimycocerosate (PDIM) biosynthesis.

P5, L218: Reference needs editing

Similarly, umaA, desA1 and 218 desA2 were also found to be up-regulated in the in vitro infection microarray analyses (Data Ref: Rohde

P5, L220 but none of the other mycolic acid biosynthesis genes that were up-regulated in vivo.

P6, L228 soon as the cultures reached hypoxia , were hypoxic (?what level of oxygen here? Be specific <1%?) the expression of the desaturases increased for ~5h, followed by a dramatic repression after ~30h of being in hypoxia.

P8: L259 reference needs editing (Data Ref: Minch et al, 2014, Minch et al, 2015).

P8: L260 Reference needs editing again and there are others within the manuscript which need correcting

2nd Revision - authors' response

31st January 2019

Reviewer #1

Minor points:

1. P4 L156: something missing from this sentence as makes no sense

We thank the reviewer for this attention to detail. We missed a period and have fixed the main text as following:

These up-regulated mycolic acid biosynthesis genes include *umaA*, *pcaA*, *desA1*, *desA2*, *fadD32* and *fabD*. In addition, genes of the operon involved in phthiocerol dimycocerosate (PDIM) biosynthesis are also up-regulated.

2. P5, L218-220: Reference needs editing.... up-regulated in the in vitro infection microarray analyses (Data Ref: Rohde

We followed the author instructions given on the submission website regarding 'Data citations' and are unsure what the reviewer is critiquing. It is possible the reviewer is unfamiliar with the data citation format of *Molecular Systems Biology*. We are seeking the editor's advice on this matter.

3. P6, L228 soon as the cultures reached hypoxia , were hypoxic (?what level of oxygen here? Be specific <1%?) the expression of the desaturases increased for ~5h, followed by a dramatic repression after ~30h of being in hypoxia.

We agree with the reviewer and have clarified the dissolved oxygen concentrations for the transient up-regulation of the desaturases.

Main text L228-230:

However, as soon as the cultures reached **complete** hypoxia (**0% dissolved oxygen**), the expression of the desaturases increased for ~5h, followed by a dramatic repression after ~30h of being in hypoxia.

4. P8: L259 reference needs editing (Data Ref: Minch et al, 2014, Minch et al, 2015).

5. P8: L260 Reference needs editing again and there are others within the manuscript which need correcting

Comments 4 and 5 are similar to comment 3. Please see response to comment 3.

Corresponding Author Name: Nitin S Baliga

Manuscript Number: MSB-18-8584